# Modelling Point Mass Balance for the Glaciers of Central European Alps using Machine Learning Techniques

Ritu Anilkumar[1,2], Rishikesh Bharti[2], Dibyajyoti Chutia[1], and Shiv Prasad Aggarwal[1]

[1]North Eastern Space Applications Centre, Department of Space, Umiam, Ri Bhoi, Meghalaya, India
[2]Department of Civil Engineering, Indian Institute of Technology, Guwahati, Assam, India

**Correspondence:** Ritu Anilkumar (ritu.anilkumar@nesac.gov.in)

**Abstract.** Glacier mass balance is typically estimated using a range of in-situ measurements, remote sensing measurements, and physical and temperature index modelling techniques. With improved data collection and access to large datasets, data-driven techniques have recently gained prominence in modelling natural processes. The most common data-driven techniques used today are linear regression models and, to some extent, non-linear machine learning models such as artificial neural networks. However, the entire host of capabilities of machine learning modelling has not been applied to glacier mass balance modelling. This study used monthly meteorological data from ERA5-Land to drive four machine learning models: random forest (ensemble tree type), gradient-boosted regressor (ensemble tree type), support vector machine (kernel type) and artificial neural networks (neural type). We also use ordinary least squares linear regression as a baseline model against which to compare the performance of the machine learning models. Further, we assess the requirement of data for each of the models and the requirement for hyperparameter tuning. Finally, the importance of each meteorological variable in the mass balance estimation for each of the models is estimated using permutation importance. All machine learning models outperform the linear regression model. The neural network model depicted a low bias, suggesting the possibility of enhanced results in the event of biased input data. However, the ensemble tree-based models, random forest and gradient-boosted regressor outperformed all other models in terms of the evaluation metrics and interpretability of the meteorological variables. The gradient-boosted regression model depicted the best coefficient of determination value of $0.713$ and a root mean squared error of $1.071$ m we. The feature importance values associated with all machine learning models suggested high importance to meteorological variables associated with ablation. This is in line with predominantly negative mass balance observations. We conclude that machine learning techniques are promising in estimating glacier mass balance and can incorporate information from more significant meteorological variables as opposed to a simplified set of variables used in temperature index models.

## 1 Introduction

We can visualize glaciers as interactive climate-response systems with their response described by changes in glacial mass over a given period (e.g. White et al., 1998). Several studies have reported the impact of climate change on glacier mass at a global and regional scale (e.g. Le Meur et al., 2007; Huss et al., 2008), with repercussions including and not limited to glacial outburst floods and diminishing water supplies. Thus, understanding the response of glacier mass balance to climate change is

crucial. Glacier mass balance is most commonly measured via (i) Direct Glaciological Method, where point measures of gain or loss of glacial ice are obtained and extrapolated for the entire glacier (e.g. Kuhn et al., 1999; Thibert et al., 2008; Pratap et al., 2016), (ii) Geodetic Method, where the change in surface elevation between two-time instances for the same portion of the glacier is estimated (e.g. Rabatel et al., 2016; Tshering and Fujita, 2016; Trantow and Herzfeld, 2016; Bash et al., 2018; Wu et al., 2018) and (iii) Indirect Remote Sensing Method, where measured mass balance is correlated with the Equilibrium Line Altitude (ELA) values or Accumulation Area Ratio (AAR) values for time series data (e.g. Braithwaite, 1984; Dobhal et al., 2021). In addition to observational data, simple temperature index-based or sophisticated physics-based energy balance models (e.g. Gabbi et al., 2014) have also been developed. Energy balance models compute all energy fluxes at the glacier surface and require measurements of input variables such as meteorological and other inputs at the glacier scale (e.g. Gerbaux et al., 2005; Sauter et al., 2020). As these models are driven by the physical laws governing energy balance, they provide reliable estimates of glacier mass balance. However, the substantial requirement for ground data to force the model, the sizeable number of parameters to calibrate and the computational complexity associated with running the model make it cumbersome to use for large areas. Temperature index models use empirical formulations between temperature and melt (e.g. Radić and Hock, 2011). The simplicity afforded by these models permits extension to large scales effectively. However, using only temperature and precipitation as inputs can lead to oversimplification. However, using only temperature and precipitation as inputs can lead to oversimplification. Further, the degree day factors (DDF) considered in temperature index models are often invariant. But studies such as Gabbi et al. (2014); Mattews and Hodgkins (2016); Ismail et al. (2023) have observed a decreasing trend in DDF, particularly in higher elevations. Ismail et al. (2023) also report the sensitivity of the DDF under the influence of the changing climate, particularly to solar radiation and albedo.

With increasing data points available, a new set of data-driven techniques has gained prominence in various domains of Earth Sciences. For e.g weather prediction (for a review, see Schultz et al., 2021), climate downscaling (e.g. Rasp et al., 2018), hydrology (e.g. Shean et al., 2020) have used data-driven models, particularly, machine learning (ML) and deep learning (DL) models. Cryospheric studies, too, have adopted the use of deep learning in several prediction problems (see review in Liu, 2021). Applications of deep learning in glaciology range from automatic glacier mapping (e.g. Lu et al., 2021; Xie et al., 2021), ice thickness measurements (e.g. Werder et al., 2020; Jouvet et al., 2021; Haq et al., 2021), calving front extraction (e.g. Zhang et al., 2019; Mohajerani et al., 2021), snow cover mapping (e.g. Nijhawan et al., 2019; Kan et al., 2018; Guo et al., 2020), snow depth extraction (e.g. Wang et al., 2020; Zhu et al., 2021), sea and river ice delineation (e.g. Chi and Kim, 2017; Li et al., 2017). The use of ML and DL in glacier mass balance estimation is significantly fewer. Initial data-driven studies used multivariate linear regression to estimate glacier mass balance from temperature and precipitation Hoinkes (1968). Subsequently, several papers have used linear regression methods for varying inputs such as temperature and pressure (Lliboutry, 1974), positive degree days, precipitation, temperature and longwave radiation (Lefauconnier and Hagen, 1990). Recent studies continue to use linear regression for modelling glacier mass balance. For example, Manciati et al. (2014) used linear regression to study the effect of local, regional and global parameters on glacier mass balance, Carturan et al. (2009) used linear regression to incorporate the effects of elevation models in the estimation of summer and winter mass balance measurements. Steiner et al. (2005) was the first to use neural networks to estimate glacier mass balance for the Echaurren glacier. Bolibar et al. (2020)

used a least absolute shrinkage and selection operator (LASSO) regression, a linear model, and a nonlinear neural network model to simulate glacier mass balance. Steiner et al. (2005); Vincent et al. (2018); Bolibar et al. (2020, 2022) are some of the few studies reporting consistently better performance of non-linear models over linear models. These studies have largely used neural networks. However, a gamut of ML techniques such as ensemble-based and kernel-based techniques exist which have largely been under-utilized for the purpose of modelling glacier mass balance. This limited utilization of ML models is potentially due to the unavailability of large ground truth datasets required for training the ML models and the perceived black-box nature of ML techniques. We aim to address this by assessing the performance of different ML models for varying training dataset sizes. Further, we aim to shed light on the interpretability of ML models by using permutation importance to explain the relative importance of the input meteorological variables. The interpretability of machine learning models is largely dependent on the input variables provided. Existing data-driven models typically use a subset of topographic and meteorological variables. For example, Hoinkes (1968) uses temperature, precipitation and cyclonic/anti-cyclonic activity, Steiner et al. (2005) uses precipitation and temperature, Masiokas et al. (2016) uses temperature, precipitation and streamflow. To the extent of the authors' knowledge, no ML-based study has attempted to use a complete set of meteorological variables associated with the energy balance equation. We expand upon this and assess the monthly contributions of each of these meteorological variables in the estimation of glacier mass balance.

Through this study, we assess the ability of ML models to estimate annual point mass balance. We use an example of each of the following classes of ML models: ensemble regression tree-based, kernel-based, neural network-based and linear models. Under ensemble regression tree-based, we chose one example of boosted and unboosted models. Specifically, we compare the performance of the random forest (RF), gradient-boosted regressor (GBR), support vector machine (SVM) and artificial neural network (ANN) models against a linear regression (LR) model. We also assess the performance for varying dataset sizes as real-world measurements are limited. Finally, to explain the role of the input features on each of the ML models, we use permutation importance described further in Altmann et al. (2010). The input features for the models are the monthly mean of 14 meteorological variables associated with the energy balance equation. We obtained the meteorological data from the ERA5-Land Reanalysis dataset (Muñoz Sabater, 2019, 2021). The target data used for training the ML models are obtained from the Fluctuations of Glaciers database (WGMS, 2021; Zemp et al., 2021) over the second-order region Alps defined by Randolph Glacier Inventory under first-order region 11: Central Europe (RGI, 2017). Section 2 of the manuscript further describes each of these datasets. In this section, we also elucidate the preprocessing steps associated with an ML approach and outline the methodology followed. In sections 3 and 4, we compare the performance of each of the models for various configurations of data availability. We also delve into the interpretability of the models from a feature importance perspective. The specific point we investigate as a part of this study can be summarized as follows:

1. Understand the utility of ML models in the estimation of glacier mass balance using limited real-world datasets

2. Identify specific use cases for different classes of ML models(ensemble tree-based, kernel based, neural network based and linear regression) pertaining to data availability, evaluation metrics and explainability

3. Investigate the ability of ML models to unravel the underlying physical processes

4. Explain the relative importance of meteorological variables contributing to the mass balance estimation on a monthly basis over the year

## 2   Data and Methods

### 2.1   Machine learning modelling

ML modelling is a data-driven set of modelling techniques. Here, we used a supervised learning framework for regression where inputs are in the form of monthly meteorological variables and targets are in the form of point measurements of glacier mass balance. The actual point mass balance measurements are the target data vital to tuning the model parameters. We do this parameter tuning by designing a loss function defining the variation between the actual mass balance measurements, i.e. the target data, and the point mass balance estimates, i.e., the model's output. We start with random initialization of model parameters and finetune the parameters to minimize the loss function. For each of the ML models used in the study, we used the mean squared error (MSE) as the loss function. Further, we obtained the features of importance by assessing permutation importance. Figure 1 depicts the complete workflow used for the study. The supplementary files include runs of such experiments that impact all the ML models in an equivalent manner.

The RF model is an ensemble-based algorithm where the base learner used is a decision (regression or classification) tree (Breiman, 2001). It relies on the principle of bootstrap aggregating or bagging (proposed by Breiman, 1996) for the generation of multiple training datasets to be used by each base learner (Dietterich, 2000). To illustrate this, assume there are $N_{data}$ samples in the training dataset $D$, and a new dataset $\hat{D}$ is generated by sampling $N_{data}$ samples with repetition. In addition to the generation of bootstrapped datasets, the decision trees are generated using a random subset of input features at every impure node of the tree instead of a complete set of features that standard regression trees use.

Like the RF model, the GBR model is an ensemble-based algorithm where aggregated base learners of decision (classification or regression) trees provide an estimate. However, it differs from the RF model because it uses boosting instead of bagging to construct ensembles. In boosting-based ensembles, base learners are typically weak learners, and the design of subsequent learners is such that the overall error reduces (Natekin and Knoll, 2013; Friedman, 2001).

The SVM model is a powerful ML tool that relies on Cover's theorem. The theorem suggests that data that might not be linearly separable in a lower dimensional space can be linearly separable when transformed into a higher dimensional space. In the context of classification, the SVM model uses a kernel to transform the data into a higher dimensional space (Cortes and Vapnik, 1995) where linear separability is feasible in the form of a hyperplane and decision boundaries. For this purpose, we use kernels such as polynomial kernel and radial basis function kernel (Vapnik, 1999). In the case of regression, the hyperplane represents the best fit line. Thus, unlike empirical risk minimization, where the difference between the actual and predicted model is optimized, the SVM model for regression uses structural risk minimization by identifying the best fit line.

McCulloch and Pitts (1943) proposed the NN models as mathematical representations of biological neuron interconnections. Hornik (1991) showed that neural networks with as few as a single hidden layer with a sufficiently large number of neurons, when used with a non-constant unbounded activation function, can function as universal function approximators. Presently,

several applications (Seidou et al., 2006; Moya Quiroga et al., 2013; Haq et al., 2014) using multiple layered NN models demonstrate that NN can infer abstract relationships between features. NN models use weighted combinations of input features in tandem with non-linearities provided by activation functions such as sigmoid, tanh and rectified linear unit (ReLU), resulting in the model output. The weights of the NN model are the model parameters obtained by optimization of the loss function.

## 2.2   Preparation of features and target data

The most crucial component in ML modelling is the availability of target data to train the model. The target data used for training should be representative of the entire population. Hence, we chose the Fluctuations of Glaciers (FoG) database (WGMS, 2021; Zemp et al., 2021) that contains measured point mass balance information (46,356 data points) globally. The study area is the Randolph Glacier Inventory (RGI) version 6 (RGI, 2017) second-order region Alps under the first-order region 11: Central Europe. This consisted of 15,727 glacier mass balance point measurements. We performed a first-level preprocessing where we considered only annual mass balance measurements (10,102 data points) and measurements from 1950 (9,595 data points) onward. We then performed an outlier removal where we considered only those points within two standard deviations of the median. This was to avoid the effects of noisy data. We finally used 9166 data points to apply our model.

The second aspect is the input features used by the model to make predictions. The network of weather stations is sparse over much of the Alpine terrain; hence, reanalysis datasets are recommended (Hersbach et al., 2020). We used the ERA5-Land reanalysis dataset (Muñoz Sabater, 2019, 2021). This data set was chosen primarily due to its comparatively high spatial resolution. This is in line with the findings of (Lin et al., 2018) and (Chen et al., 2021) that suggest that datasets with higher spatial resolution effectively represent the orographic drag and mountain valley circulation which in turn results in improved performance for orographically complex terrain. The choice of variables reflected the contribution of the same to the energy balance equation that drives mass balance modelling from a physical standpoint. We considered the following fourteen variables for the modelling: the temperature at 2m, snow density, snow temperature, surface net solar radiation, total precipitation, forecast albedo, surface pressure, surface net solar radiation downwards, snowfall, surface net thermal radiation, snowmelt, surface sensible heat flux, snow depth and surface latent heat flux (For details, see Muñoz Sabater et al., 2021). We consider these meteorological variables because of their effect and representation of the accumulation and ablation process and define the variables expected to represent accumulation processes as accumulation variables (e.g. snowfall, forecast albedo) and melt processes as ablation variables (e.g. temperature, solar radiation). The monthly mean of each of the accumulation and ablation variables was considered. Thus, we have 168 total input parameters. For each of these variables, we extracted the data using the nearest neighbour algorithm, using latitude, longitude and year of the glacier mass balance measurement from the FoG database. Thus the final dataset has 168 input features and 9166 data points.

We then normalised the data points using a min-max scaling to ensure the absence of user-conceived bias in the model. We have split the dataset using a random split where 70% of the total dataset is used for training the model and 30% is used for testing the model performance. The training split is used in a 3-fold cross-validation process for tuning the hyperparameters as described further in Section 2.3. Finally, we rescaled the model's predictions to assess the model metrics, such as root

mean squared error (RMSE), mean absolute error (MAE) and normalized mean squared error (nRMSE) and normalized mean absolute error (nMAE) in the measured point mass balance units.

## 2.3 Hyperparameter Selection and Finetuning

In typical ML workflows, we split the complete dataset (set of features and target data) into training, validation and testing. We fit the model to the data using the training subset, tune the hyperparameters using the validation subset, and report the independent performance metrics using the testing subset. In our case, we used a 70%-30% split for training and testing. We have considered a hyperparameter grid with all combinations of values that each hyperparameter can take (see Tab. 1). Rather than using a fixed ratio subset for validation as was the case with the testing, we divided the training data subset into three equal folds. Two folds are randomly selected as the training set and the third fold is used for validation. The validation score is noted and the process is then repeated for the other fold combinations. The mean validation score for each hyperparameter setting obtained from the grid is used for the selection of the optimal hyperparameters. We compute the validation score as the negative of the RMSE after scaling the target data to a range between 0 and 1. Thus a more negative validation score results in a more significant error.

For the RF model, we tuned the number of trees. We maintained the maximum depth as indefinite, leading to tree expansion until all nodes were pure. We considered all features to obtain the best split, ensuring minimum bias. As computation for absolute error is slow at each split, we used the squared error as the splitting criterion. This ensured the minimisation of the variance after each split. For the GBR model, we tuned the number of trees, maximum depth of each tree (which affects the randomness in the choice of features in each tree), and subsampling ratio (for stochastic gradient boosting). Larger values of maximum depth, such as the indeterminate depth of the RF model, are not used as GBR functions with weak learners to increase the randomness. The SVM model hyperparameter finetuning involved kernel selection and a choice of the regularisation parameter. Further, in the case of polynomial kernels, the degree of the polynomial was also tuned. For the NN model, we used a fully connected feedforward network where the hyperparameters of the number of layers and number of neurons in a layer were tuned. The activation function ReLU was used to incorporate non-linearity. We used the adam (Kingma and Ba, 2014) optimiser to minimise the loss function. The training process was performed for 500 iterations with early stopping in the event of convergence before completing the iterations. The NN models for each set of hyperparameters converged before the completion of the 500 iterations.

## 2.4 Performance Evaluation

The testing dataset evaluation metrics used to assess the models' performances are the coefficient of determination ($R^2$) which represents the percentage deviation between the target and model predictions, $RMSE$ which represents the absolute deviations between the target and the model predictions. Lower $R^2$ values suggest that the model does not represent the targets well. Values close to one indicate a strong linear correlation. Lower $RMSE$ values are preferable as this quantifies the variance between the targets and predicted values. Additionally, we report the slope and additive bias using reduced major axis (RMA) regression.

We used RMA regression slope and bias to ensure symmetry about the $y = 1$ line. This is preferable as there exist uncertainties in both target data and outputs.

ML models are heavily reliant on the availability of training data. To understand the effect of data availability on the model performance, we perform an experiment on varying the training sizes. We split the original dataset into subsets of iteratively increasing sizes. We partition each subset into training and testing partitions using a 70:30 ratio. For each subset, we train all the models using the training partition and computed the evaluation metrics over the testing partition.

## 2.5  Feature Importance

The feature importance is represented using permutation importance described in Altmann et al. (2010). Here, we disregard individual features from the model at each iteration and recorded the reduction in evaluation score. This is repeated for each input feature. We normalize the obtained permutation importance for each model and express the importance of each input meteorological variable as a percentage. A comparative analysis of the obtained feature importance is performed on two counts: (a) Features that are most important. Here the most important 10% of the features are considered. Thus the 17 most important meteorological variables out of 168 used are reported. This is represented in Supplementary material S1. (b) Percentage importance associated with the accumulation months (November to March) and the ablation months (June-September) is summed and graphically represented for each model in Fig. 6.

## 3  Results

This section describes the major outcomes of the study categorized as the role of dataset size for the effective training of each ML model (see Fig 2), the performance and feature importance associated with each ML model. Figure 3 represents the comparative performance of each of the models in terms of the accuracy metrics $RMSE$, $R^2$, Slope and Additive Bias. A scatter plot of modelled point mass balance and target data is represented in Fig 4. Figures 5 (a), (b), (c) and (d) represent the hyperparameter tuning associated with the models. The feature importance for all input variables summed over the ablation and accumulation months is represented in Fig 6. The most important meteorological variables (10% of total number of variables) associated with each model are represented in Supplementary material S1.

## 3.1  Role of Training Dataset Size

The number of samples required for training the ML models depends upon the complexity of the model. Thus each of the models used in this study is variably sensible to the number of training samples. We use the evaluation metrics of RMSE and correlation coefficient to assess the requirement of training samples for each of the models. Figure 2 depicts the training and testing metrics varying with the size of the training dataset. The training metrics do not show significant change after 20-30% of the training dataset size for the LR, RF, GBR and SVM models and after 40% for the NN model. This illustrates the larger number of trainable parameters resulting in the requirement of larger datasets for artificial neural networks for training. The

testing performance of each of the models do not show significant change for training dataset sizes larger than 50%. We observe that while a downward trend is evident with the addition of new data, the rate of improvement is slower.

It is interesting to note that RF, GBR and LR models see an increase in training $MAE$ as opposed to a consistent decrease in testing $MAE$ with increasing training samples. This depicts the tendency of these models to overfit the training samples in the case of smaller datasets. This is evident when observing the order of variation in the training and testing evaluation metric for smaller datasets. E.g. GBR depicts a training $MAE$ of 357 mm we and a testing $MAE$ of 1183 mm we at 10% training dataset size and training $MAE$ of 659 mm we and a testing $MAE$ of 774 mm we at 100% training dataset size. Thus, care must be taken when using RF and GBR for smaller datasets as they are susceptible to overfitting. The performance of the LR model deteriorates for training, and testing performance is also poor. This is not due to overfitting but due to the inability of the model to explain the complex relationship between the inputs and the target. NN requires larger datasets for the training of the model. Figure 2b depicts the superior performance of RF, GBR, and SVM in the event of limited dataset availability. However, we have seen that RF and GBR show a marked increase in training MAE with increasing training samples which suggests overfitting to limited datasets. Thus SVM is more robust to smaller datasets.

## 3.2 Performance of RF modelling

The best-performing RF model resulted in a testing $RMSE$ value of 1083 mm we and an $R^2$ value of 0.71. The testing $MAE$ value is 782 mm we and the testing $nRMSE$ and $nMAE$ are 0.55 and 0.40 respectively. The training $RMSE$ value is 934 mm we, $MAE$ value is 672 mm we, $nRMSE$ is 0.48, $nMAE$ is 0.34 and $R^2$ value is 0.80. We observe that hyperparameter tuning is not important, and no major variations were observed upon changing the number of estimators. The slope of RF was closest to 1 with a value of 0.752 for the training samples and 0.744 for the testing samples. Both training and testing additive bias were negative, suggesting the model underestimated point mass balance (Fig 3).

Feature importance analysis using permutation importance considering the 17 ( 10% of all features) most essential features indicates the RF model is highly influenced by Downward Solar Radiation in January, Net solar radiation for July, Downward thermal radiation in June, Temperature at 2m in June, forecast albedo in February and December, Snow Depth in January and July, snow density and snowmelt in July, sensible heat flux in December, January, March and May, latent heat flux in August and Surface Pressure in June and July. Permutation importance for the RF model summed over the accumulation months highest importance scores for sensible heat flux followed by downward solar radiation and forecast albedo. Each of these variables depict a summed percentage importance between 6-9%. Snow depth and pressure are also important with a summed percentage importance between 3-6%. For the ablation months, only pressure is observed to have a summed percentage importance greater than 6%. Sensible heat flux, net solar radiation, latent heat flux, snow depth, forecast albedo, snow density and temperature at 2m display summed percentage importance between 3-6%.

## 3.3 Performance of GBR modelling

Tuning the maximum depth permitted for each weak learner tree was important in estimating the best model, and varying the number of weak learner trees during hyperparameter tuning improved performance in the case of smaller depths of the

weak learners. Deeper tree structures did not significantly change the model's performance upon changing the number of trees. Stochastic gradient boosting (subsampling at 0.7) resulted in reduced performance. The hyperparameter combination of the best performing GBR model is 100 trees with a maximum depth of 5 nodes (Fig 5 (a)). The best performing GBR model resulted in a testing $RMSE$ value of 1071 mm we and an $R^2$ value of 0.71. The testing $MAE$ value is 774 mm we and the testing $nRMSE$ and $nMAE$ are 0.55 and 0.39 respectively. The training $RMSE$ value is 759 mm we, $MAE$ value is 659 mm we, $nRMSE$ is 0.39, $nMAE$ is 0.34 and $R^2$ value is 0.80.

The most important meteorological inputs for the GBR model are Snowfall in July, Downward solar radiation in January and December, Forecast Albedo in December, January, February, March and May, Sensible Heat Flux in January, March, May, November and December, Temperature at 2m in June and August, snow depth in June and surface pressure in August. Note the marked importance associated with ablation meteorological variables and the months associated with ablation. Permutation importance expressed as a percentage and summed over the accumulation months depicts the most importance to forecast albedo followed by sensible heat flux, with both variables depicting a summed percentage importance greater than 10%. Among other meteorological variables, downward solar radiation, net solar radiation and snow depth in the accumulation months are also important. The ablation months depict higher summed importance values with forecast albedo in these months prominent. Sensible heat flux, latent heat flux, surface pressure, snowfall, snow depth and temperature at 2m above the surface are also important.

### 3.4 Performance of SVM modelling

The SVM model depicted large fluctuations in the validation score with changes in the hyperparameters. This is represented in Fig 5 (b). We considered the hyperparameters of the kernel, degree (for polynomial kernel) and regularisation (penalty) factor. The sigmoid kernel resulted in evaluation metrics markedly poorer than the radial basis function (RBF) kernel and polynomial kernels. The sigmoid kernel was excluded from the graphical representation of the validation score to emphasise the variations observed in the other kernels. The polynomial kernel at larger degrees consistently performed better than the RBF kernel in the case of regularisation tuning lower than 1. For larger regularisation parameters, the RBF kernels demonstrated better performance. The best-performing model in this study is the RBF kernel (penalty factor: 10.0). Figure 5 (b) depicts the results of hyperparameter tuning for the SVM kernel. The testing $RMSE$ values for the model are 1085 mm we and $R^2$ value is 0.70. The testing $MAE$ value is 836 mm we and the testing $nRMSE$ and $nMAE$ are 0.56 and 0.43 respectively. The training $RMSE$ value is 727 mm we, $MAE$ value is 727 mm we, $nRMSE$ is 0.37, $nMAE$ is 0.37 and $R^2$ value is 0.76.

The permutation importance associated with Sensible Heat Flux March is most important, as is the sensible heat flux associated with April, May, June and December. Latent heat flux in August and October is important. Snowfall in October and snow density for the months of November, December and January are important. The temperature at 2m above the surface in June and July, downward solar radiation in December and forecast albedo in August, October and December are important. Summing the percentage importance over the accumulation and ablation months, we observe that sensible heat flux in the accumulation months is most important, followed by snow density and downward solar radiation. These three variables depict a summed percentage importance of more than 6%. The temperature at 2m above the ground and forecast albedo de-

pict importance between 3-6% for the accumulation months. For the ablation months, sensible heat flux continues to depict a summed percentage importance of more than 6%. Latent heat flux, snow density, forecast albedo and temperature at 2m above the surface also depict a summed percentage importance between 3-6%.

### 3.5 Performance of NN modelling

The NN model performance is highly susceptible to hyperparameter selection. We varied the number of hidden layers in the network and the number of neurons in each hidden layer. Figure 5 (c) and (d) depicts the variation in performance of the model for each of these cases. On the left is the variation in the number of neurons for a single hidden layer. A larger number of hidden neurons permits more combinations of the inputs that can affect the targets. The improved performance with the increasing size of neurons illustrates the role of the complexity of the model in estimating mass balance. Increasing the number of layers also affects the performance of the NN model, with the best performance obtained using two hidden layers. This further emphasises the importance of incorporating non-linear elements in estimating point mass balance. A larger number of hidden layers did not significantly improve performance as the larger number of parameters demanded a larger training dataset to avoid overfitting and to complete the training. The testing $RMSE$ values for the best-performing model are 1096 mm we and $R^2$ value is 0.70. The testing $MAE$ value is 836 mm we and the testing $nRMSE$ and $nMAE$ are 0.56 and 0.43 respectively. The training $RMSE$ value is 773 mm we, $MAE$ value is 773 mm we, $nRMSE$ is 0.39, $nMAE$ is 0.39 and $R^2$ value is 0.76.

The most important meteorological variables in terms of the percentage permutation importance for the NN model are the Sensible Heat Flux for March, April and May, Latent Heat Flux in July, Surface Pressure in February, the Net Solar Radiation in May and September, downward solar radiation in December and forecast albedo in July. Snow Density in December and the snow depth January, February, April, July, September, October and December are important. We see that snow depth across the year dominates the important meteorological inputs for this model. Upon summing the percentage importance for the accumulation and ablation months, we observe that snow depth is the most important for both accumulation and ablation months. Snow density, pressure, sensible heat flux and downward solar radiation are also important in the accumulation months, with a summed percentage importance value between 3-6%. For the ablation months, net solar radiation is also important. Snow density, forecast albedo, latent heat flux and sensible heat flux are also important, with summed percentage importance values between 3-6%.

### 3.6 Performance of LR modelling

The testing $RMSE$ values for the LR model are 1248 mm we and $R^2$ value is 0.58 and the training $RMSE$ values are 1197 mm we and $R^2$ value is 0.61 (Fig 3). The testing $MAE$ value is 941 mm we and the $nRMSE$ and $nMAE$ are 0.64 and 0.48 respectively. The training $MAE$ value is 935 mm we, $nRMSE$ is 0.61 and $nMAE$ is 0.48.

Snow depth over most of the year is the most important feature for the model, with surface pressure also playing an important role. Other features do not depict as high an importance value. However, relative importance varies across the months.

 # 4 Discussion

## 4.1 Comparison of Model Performance and Associated Errors

The performance of each of the models was evaluated using an independent test dataset. The GBR model resulted in the best testing performance MAE, RMSE and $R^2$ values outperforming the RF model, SVM and NN models. Neural networks resulted in better bias performance. RF, GBR, SVM and NN significantly improve upon the LR model's metrics. The ability of all non-linear models to outperform the linear model is further depicted in each model's scatter plot (Fig. 4). This is in agreement with similar studies in other domains, such as King et al. (2020) who showed that tree-based models such as RF were preferable to LR models for the bias-correction of snow water equivalent and Rasouli et al. (2012) who depicted the efficacy of non-linear models in estimation of streamflow when compared to linear models.

The performance of all models is affected by the uncertainties associated with the input features and targets. Inherent errors exist in point mass balance estimates as heterogeneity is not captured sufficiently by the available measurements (Zemp et al., 2013; Van Tricht et al., 2021). Of the 727 locations with uncertainty estimation performed, we note a mean uncertainty of 62 mm we, which can adversely impact performance evaluation. The uncertainty estimates for the remaining point locations are unknown; hence, their impact is not constrained. In this study, we did not consider the effect of topography and debris cover for the models. This can lead to inflated RMSE values.

Further, the use of input meteorological reanalysis data can result in bias, especially in locations without sufficient ground stations (Zandler et al., 2019; Guidicelli et al., 2022). Specifically for the use of ERA5 Land data in complex terrain, Wu et al. (2023) reports that while ERA5 Land represents the intra-annual variations in precipitation characteristics, there is a positive bias in the precipitation variables. Similarly, in the case of temperature, Zhao and He (2022) show through correlation and RMSE analysis that while the ERA5 Land dataset captures the temperature trends effectively, the magnitude of the values is not well represented. Thus, we suggest using a bias correction step such as that proposed by Cucchi et al. (2020) in the case of RF, GBR and SVM models. Moreover, the reanalysis data do not fully reflect point scale data as it has a coarse resolution. (Lin et al., 2018) depicts the impact of resolution in simulating drivers of local weather in complex terrain and shows that coarser resolutions do not account for orographic drag. Approaches such as using a scaling factor or lapse rates have been attempted by studies (e.g. Radić et al., 2014; Maussion et al., 2019). However, these studies largely utilize precipitation and temperature as inputs, the scaling of which with elevation is fairly straightforward. Choosing appropriate scaling factors for other meteorological variables that drive glacier mass balance (e.g sensible and latent heat fluxes, albedo) is not intuitive. We note that the effects of the larger scale of the input variable will persist in the model. However, these effects will be consistent across all the models. Thus the effect of the input variable scale is represented by the uncertainty of all models, and a relative analysis of the performance of models will remain well-founded.

## 4.2 Role of Training Dataset Availability

The testing performance improves by increasing the number of training samples. We observe that for a larger number of data points, marginal improvement is observed upon increasing the number of samples further. The reduction in the rate of

improvement for all models suggests that all models have been successfully trained. However, the marginal improvements observed suggest a potential improvement in model performance is possible when including more data samples. The RF and GBR models overfit the training samples in the case of smaller datasets. The NN model training and testing metrics depict improved performance with training size. The NN model had the most trainable parameters and hence is the most data-intensive. A larger number of training samples is essential for models with a larger number of trainable parameters. The training performance of the LR model deteriorates with increasing training samples. While the graph (LR model of Fig. 2) appears similar to the RF and GBR training graphs, the relatively close training and testing metrics values suggest that overfitting is not the likely cause. Rather, it suggests that the model cannot explain the non-linear relationship between the inputs and the target.

Further, Fig. 2 represents each model's variation in training and testing evaluation metrics. Each model was trained and tested over each dataset size. For each model, the box plots are generated utilising the outcome of the models developed using varying training dataset sizes. The training performance, as expected, is better than the testing performance as the model parameters are tuned to fit this dataset. The range of values is more extensive for the testing errors as a result of overfitting in the case of smaller datasets. In such cases, the use of the SVM model yields better results.

## 4.3  Unraveling the Physics using Machine Learning-Derived Feature Importance

Assuming a winter accumulation-type glacier, we expect the months of November to March to be dominated by accumulation processes and June to September to be dominated by ablation processes. Analysis of the permutation importance (by percentage) of the features of each model was studied month-wise based on a physical understanding of which season-specific features will be most important. Figure 6 represents the summed feature importance for each input variable in the accumulation and ablation months. We sum the percentage importance rather than the feature importance values to permit comparison between models. We expect temperature (2m) for ablation seasons to be significant compared to temperatures in the accumulation season. This is not well reflected when using the LR model. While all the ML models show the reduced importance of temperature in the accumulation months, it is most pronounced in the case of the RF and GBR models. A similar trend is expected for the downward thermal radiation and snowmelt. Here, too the LR model does not reflect the expected outcome. All ML models depict reduced importance in the accumulation months, with a pronounced reduction observed in the RF and GBR models. In the case of snowmelt, all ML models and the LR model follow the expected response. Snow depth throughout the year is important when considering snow density. We expect the depth in the ablation months to be important. All models portray this except the SVM model. We observe that the LR model relies heavily on snow depth to estimate the mass balance. The SVM model reports the exaggerated importance of snow density in the accumulation months. While we expect more importance to precipitation terms such as total precipitation and snowfall in the accumulation months, we do not observe this for any model. The LR model did show a weak reduction in the importance of total precipitation and snowfall. However, the ML models showed only a weak reduction or a weak increase in importance. This is possibly a result of the scale of the meteorological variables used not sufficiently representing the influence of orographic water vapour transport that results in precipitation (Lin et al., 2018; Chen et al., 2021).

Net solar radiation and albedo are important ablation components. Albedo over snow-covered regions is higher than that of exposed ice or firn. At higher elevations and in summer months, we expect lower albedo values. Thus variations in albedo are of significance. In the case of ERA5 Land, the forecast albedo variable represents both the direct and diffuse radiation incident on the surface with values dependent on the land cover type. It is calculated using a weight applied to the albedo in the UV-visible and infrared spectral regions. The albedo of snow and ice land covers differs in the UV-visible and infrared spectral regions. This makes forecast albedo more important than broadband albedo, which depends only on the surface net solar radiation and the surface solar radiation downwards. The expected importance of the albedo is observed in the RF, GBR, NN and SVM model. LR models, in contrast, depict very low importance of albedo for the accumulation months. Thus we see that the ML models well represent the importance of the ablation features. This is in agreement with the predominantly negative mass balance observed in in-situ measurements.

We can observe that the importance associated with the meteorological variables is not dominated solely by total precipitation and temperature, as with temperature index models. Thus, ML modelling can represent the contributions of a complete set of variables with lesser complexity and ease of use than physical models. This also emphasises the requirement for ML models to use all meteorological variables of interest, as opposed to a subset of them. This is the case with studies such as Bolibar et al. (2020). Further, our results agree with the studies conducted by Steiner et al. (2005) and Bolibar et al. (2022) in that artificial neural networks capture the complexity of the mass balance estimation using non-linear relationships between inputs. However, we propose that other ML models, notably ensemble tree-based methods, can be used for equivalent to improved estimates in case of fewer real-world data samples for training. This has also been observed in other studies (e.g. Bair et al., 2018) For this case, feature importance derived using permutation importance for the ensemble-based models, RF and GBR, represented the expected role of meteorological variables in determining feature importance. The evaluation metrics also emphasise the performance of these models.

### 4.4 Relevance to future studies

With the emergence of artificial intelligence techniques, a number of studies have employed deep learning algorithms for numerous applications. A majority of these studies use neural networks to incorporate non-linearity in the modelling of various Earth observation applications. However, a host of ML techniques exist which remain under-utilized. This is being studied in the ML community (e.g. Fernández-Delgado et al., 2014, studied 179 classification models) and it has been observed that for tabular datasets, tree-based models remain state of the art (Shwartz-Ziv and Armon, 2021; Grinsztajn et al., 2022) for both classification and regression problems for medium-sized datasets (training samples under 10,000). Our study also depicts the improved performance of GBR models, which aligns with these recent findings. While it largely follows the assumptions made by Grinsztajn et al. (2022), we demonstrate the case of regression with heterogeneous and interdependent input features and a voided assumption of the identical and independent distribution of samples also depict a better performance by ensemble tree-based models. Glacier mass balance datasets being typically medium-sized datasets with correlated input features, we recommend that studies aiming to use ML for modelling the Earth system consider the ensemble-based techniques. Many ensemble-based techniques exist, including bagging as used by RF and boosting as used by Adaboost and GBR. Further,

studies that combine ensemble trees models with deep learning are also being used effectively (e.g. Shwartz-Ziv and Armon, 2021, used XGboost in tandem with an ensemble of deep models). Bolibar et al. (2020) utilize a Leave-One-Year-Out and Leave-One-Glacier-Out mode of testing the performance of the model. This is in line with Roberts et al. (2017) who suggest that spatially and temporally structured datasets would benefit from a manually designed blocking strategy. As the testing and validation splits will result in similar effects in all the models, performing the grouped splitting does not provide immense value to this study. However, for cases where a single model is to be used to estimate glacier mass balance, the Leave One Glacier Out and Leave One Year Out techniques are useful.

An aspect not considered in this study is a transfer learning approach to the ML modelling where glacier mass balance datasets from other locations can be used to pre-train the neural network and generate an initialization of weights to be tuned by the dataset of the region of interest (see Anilkumar et al., 2022). In line with utilizing datasets from other locations, another aspect to consider with glacier mass balance datasets is the generalizability of the models. Understanding which machine learning model can be used for local, regional and global analysis is important and will be a useful study to take up. Feature importance associated with the local, regional and global analysis also will provide new insights into the changes in the glacier mass balance at these scales. An important factor to note is that through this study, we have considered annual mass balance measurements as opposed to seasonal measurements due to the paucity of sufficient datasets to train a multi-parameters machine learning model fully. The role of ablation and accumulation variables will be better represented in the case of seasonal measurements and is an avenue to explore through future studies.

## 5   Conclusions

In this study, we constructed 4 ML models to estimate point glacier mass balance for the RGI order one region 11: Central Europe. We used the ERA5-Land reanalysis meteorological data to train the models against point measurements of glacier mass balance obtained from the FoG database. In addition to the NN model, which is being increasingly utilised for glacier mass balance estimation, we used other classes of ML models, such as ensemble tree-based models: RF and GBR, and the kernel-based model: SVM. We compared these ML models with an LR model commonly used for mass balance modelling. Care must be taken to tune the hyperparameters for the GBR, NN and SVM models. We observe that for these models, hyperparameter tuning was beneficial for improving the estimates of glacier mass balance. For smaller datasets, ensemble models such as RF and GBR depict overfitting. The NN model requires more data samples for effective training. The SVM model can effectively be used in the case of a smaller number of data samples, which is characteristic of real-world datasets. The LR model is consistently unable to capture the complexity of the data and underperforms. For larger datasets, ensemble models such as RF and GBR perform slightly better in terms of $R^2$ and $RMSE$. However, NN models depict the least bias. The meteorological variables obtained from reanalysis datasets are associated with high bias. Using NN and LR models permits us to use them directly. For other models, bias correction should be incorporated in the preprocessing. Representation of real-world features is also performed more effectively by RF and GBR models. These models indicate the importance of ablation features dominating the mass balance estimates. This is expected as the mass balance measurements are primarily negative.

Further, feature importance suggests that features such as forecast albedo, sensible heat flux, latent heat flux and net solar radiation also play a pivotal role in estimating point mass balance. Thus inclusion of these additional variables might be of importance for future studies.

*Code and data availability.* The data used for the study is the monthly mean ERA5-Land reanalysis product for inputs features and point mass balance measurements from the Fluctuation of Glaciers database for the target data. The code for processing the data and applying all models used in this study is available at https://github.com/RituAnilkumar/pt-gmb-ml

*Author contributions.* RA, RB and DJC were involved in the design of the study. RA wrote the code for the study and produced the figures, tables and first draft of the manuscript using inputs from all authors. RB, DC and SPA proofread and edited the manuscript. RA performed the first level of analysis, which was augmented by inputs from RB, DJC and SPA.

*Competing interests.* The authors report that this study contains no competing interests

*Acknowledgements.* We acknowledge the contribution of the journal editors, particularly Dr Emily Collier, for the thorough manuscript handling. We thank Dr Jordi Bolibar and the anonymous reviewer whose detailed suggestions and inputs have substantially improved the quality of the manuscript. We also acknowledge the engaging discussions with peers, most notably Dr Aniket Chakraborty, who always lent a patient ear and sound suggestions to roadblocks along the way.

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

**Table 1.** Grid of settings used for hyperparameter tuning of each of the models

| Machine learning model | Hyperparameter | Values |
|---|---|---|
| Random Forest | Number of trees | 10,20,50,100 |
| Gradient Boosted Regressor | Number of trees | 50,100,200 |
| | Subsampling | 0.7, 1.0 |
| | Maximum Depth | 3,5,10 |
| Support Vector Machine | Cost | 0.1, 1, 10, 20 |
| | Kernels | Sigmoid, Radial Basis Function, Polynomial |
| | Degree (polynomial kernel) | 2, 3, 4, 5 |
| Artificial Neural Network | Number of layers and nodes | **1:** 10, 50, 100, 200, 300, 400, 500, |
| | | **2:** (100, 50), (200, 100), (400, 200), (200, 400) |
| | | **3:** (400, 200, 100), (500, 200, 100), (200, 100, 50), (100, 50, 10), |
| | | **4:** (200, 300, 400, 500), (300, 200, 100, 50), (200, 100, 50, 10) |

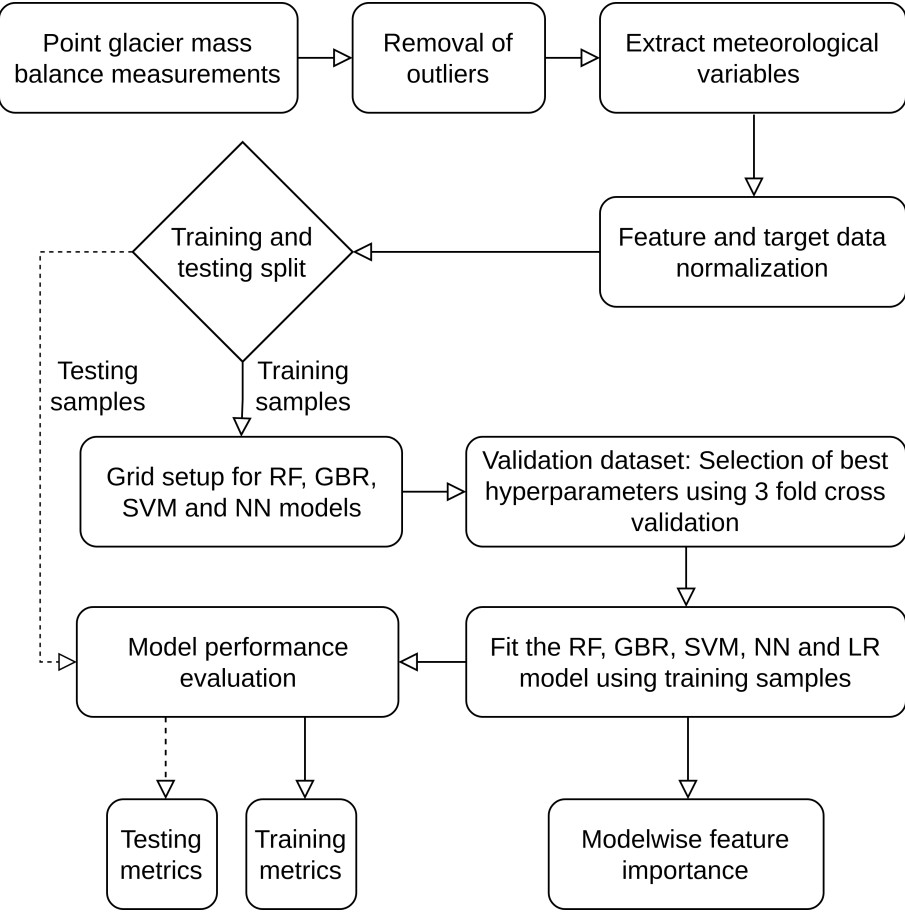

**Figure 1.** Flowchart of the methodology

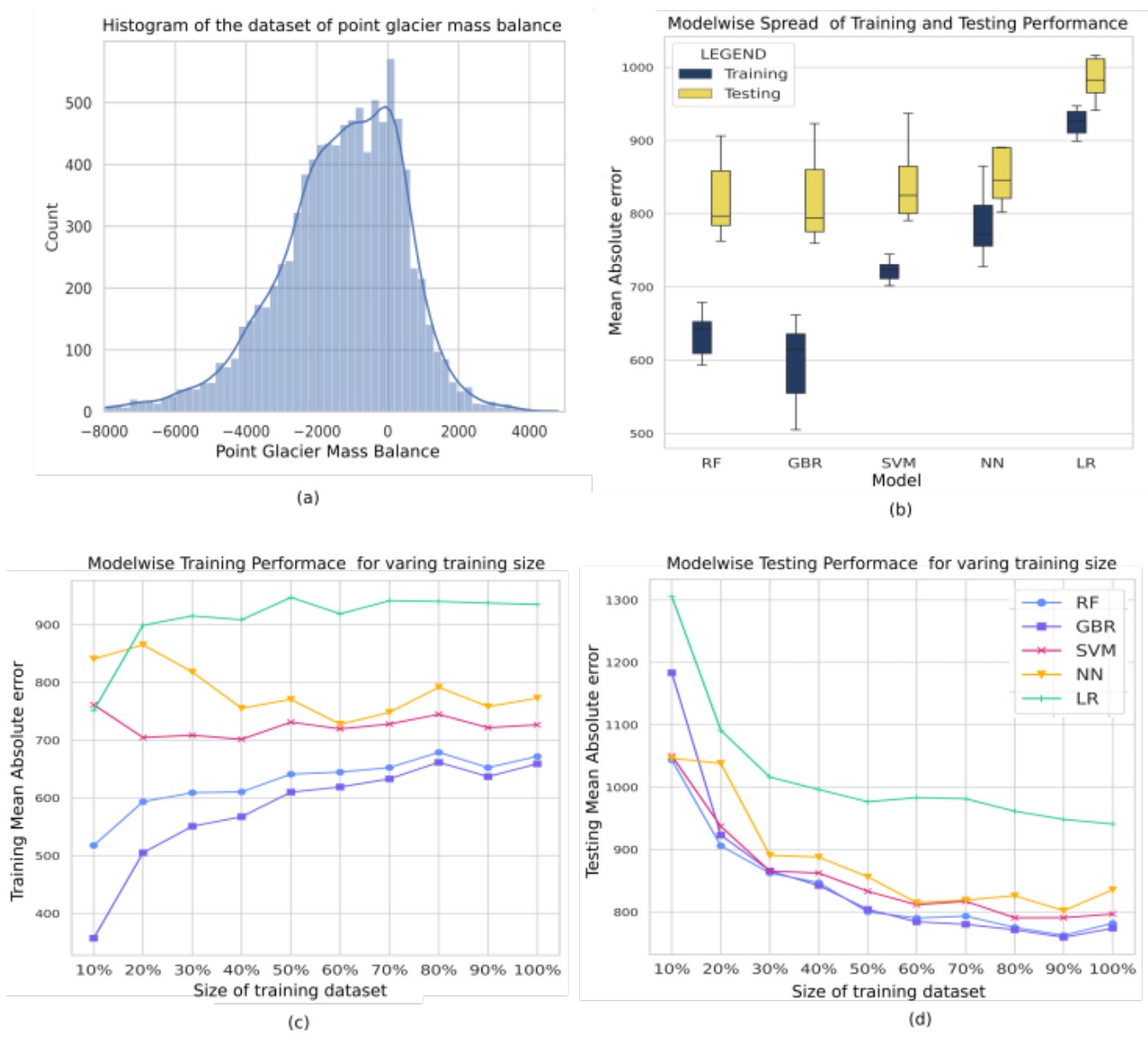

**Figure 2.** (a) Histogram depicting the distribution of the glacier mass balance measurements used for the study. (b) Box-and-whisker plot depicting the training and testing $MAE$ (in mm we) and $r$ values for varying the size of the training dataset for each of the models. The box represents the quartiles 1 to 3 and the whiskers represent the rest of the distribution ignoring outliers. (c) Modelwise training mean absolute error (in mm we) for varying the size of the training dataset size. (d) Modelwise testing mean absolute error (in mm we) for varying the size of the training dataset size. Note, the training dataset size is expressed as a percentage of the largest size of the training dataset i.e. 6416 data points

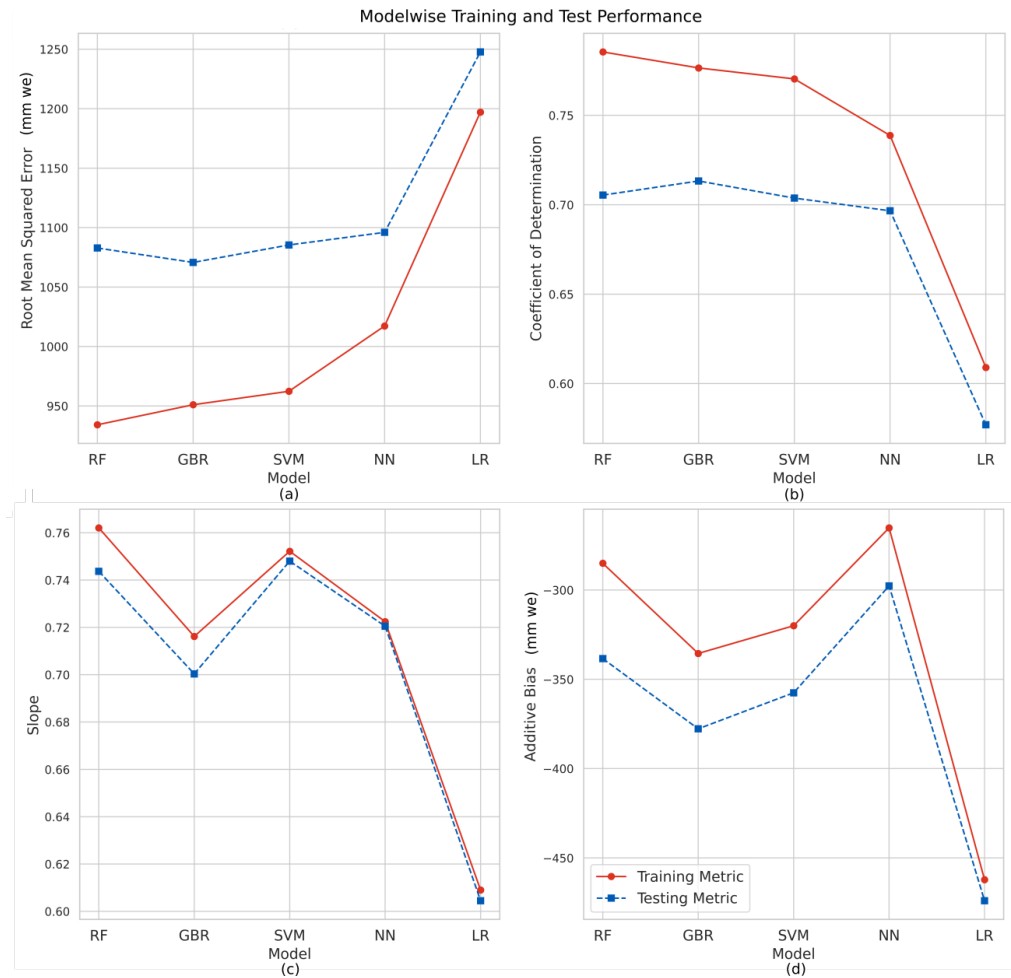

**Figure 3.** Training and testing performance of each of the models: Random Forest (RF), Gradient Boosted Regression (GBR), Support Vector Machine (SVM), Artificial Neural Network (ANN) and Linear Regression (LR) depicted using the performance metrics (a) Root Mean Squared Error, (b) Coefficient of Determination, (c) Slope and (d) Additive Bias

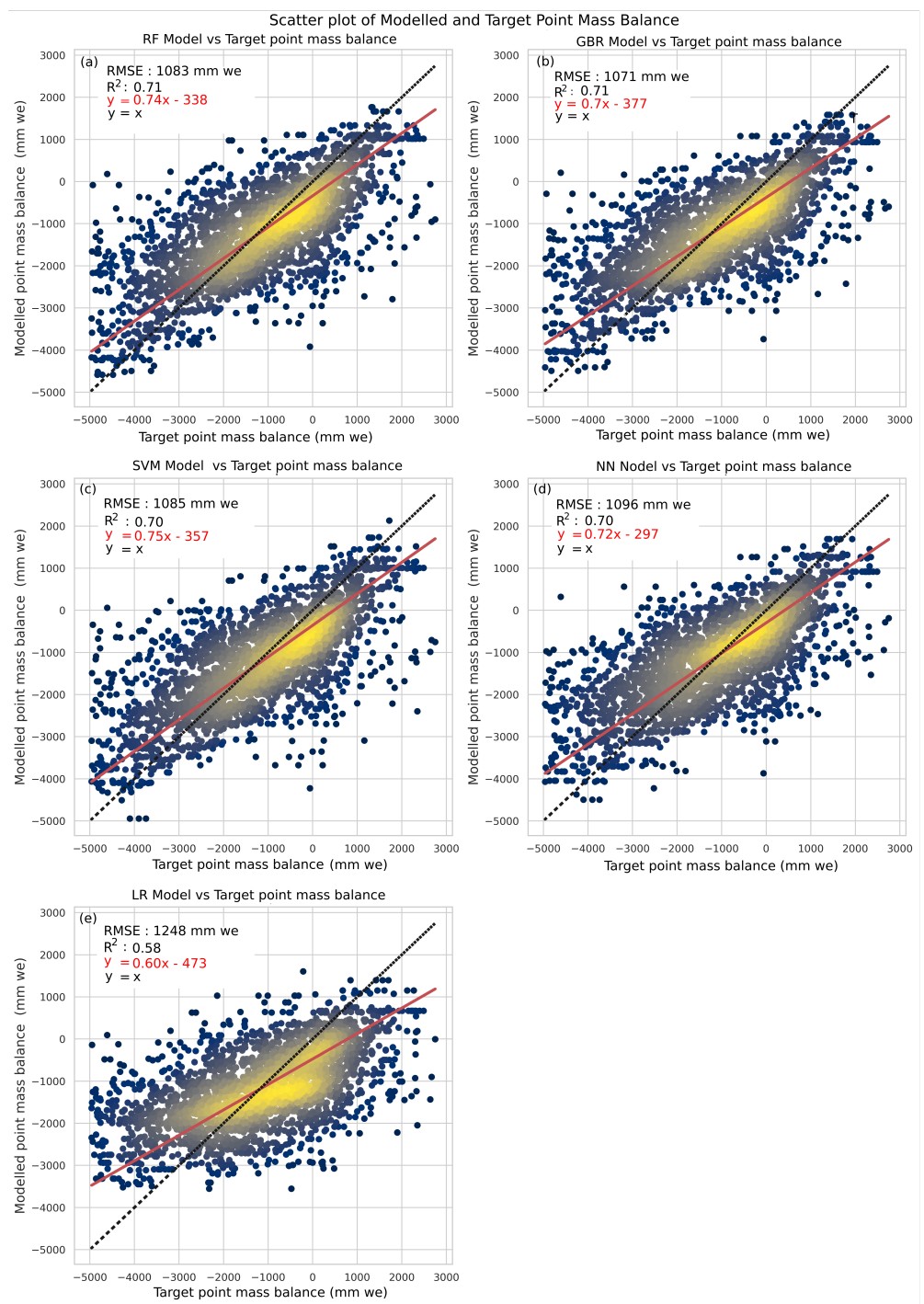

**Figure 4.** Testing scatter plot depicting the performance for each of the models: Random Forest (RF), Gradient Boosted Regression (GBR), Support Vector Machine (SVM), Artificial Neural Network (ANN) and Linear Regression (LR)

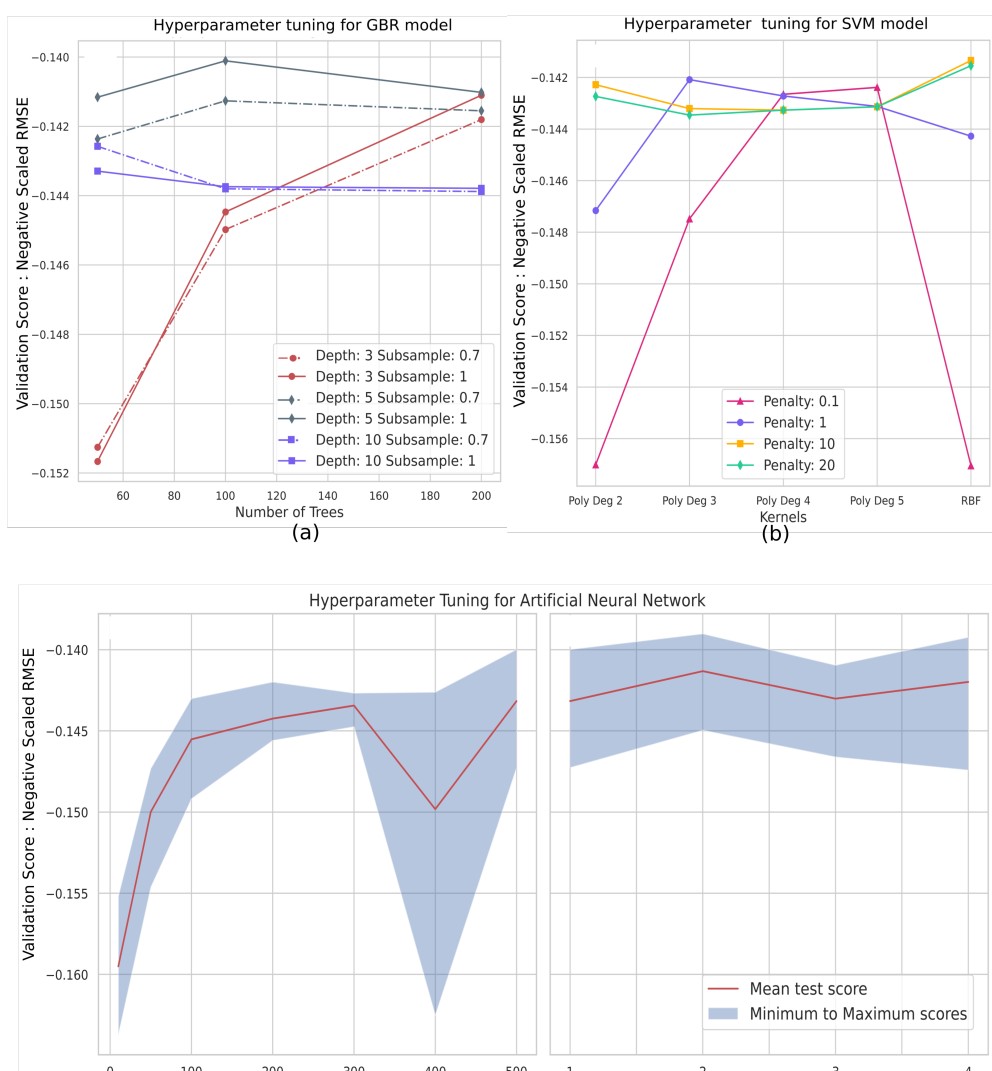

**Figure 5.** Hyperparameter tuning for the (a) GBR model varying the number of trees, maximum depth of each tree and subsampling fraction, (b) SVM model varying the penalty parameter and kernel as well as degree in case of the polynomial kernel, (c) NN model varying the number of neurons in a single hidden layer and (d) NN model varying the number of hidden layers. The validation score used is the negative scaled RMSE which is the negative of the normalized RMSE values that can easily be used to rank the hyperparameter settings

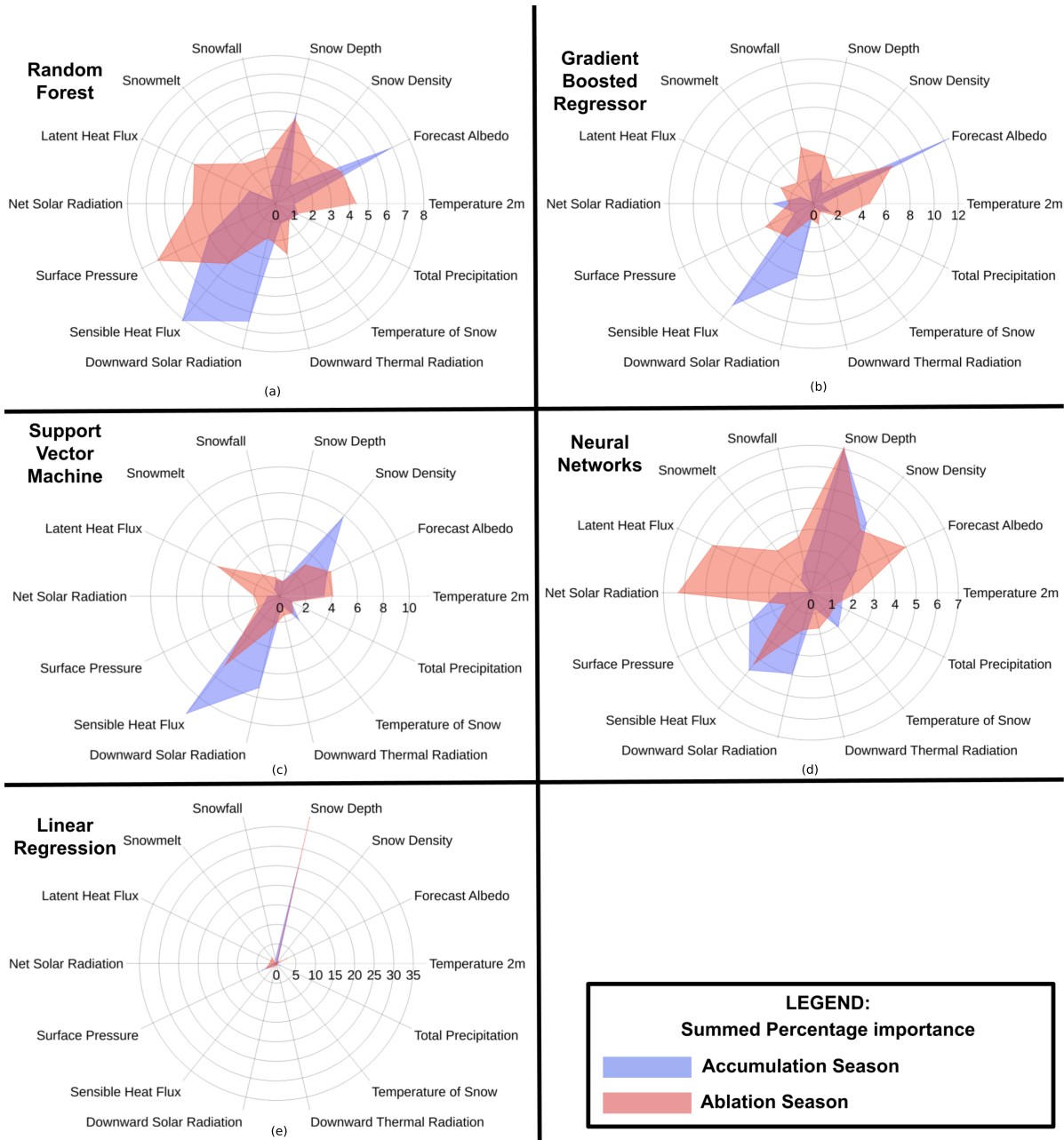

**Figure 6.** Percentage importance of all features summed over the accumulation and ablation season for the models: Random Forest (RF), Gradient Boosted Regression (GBR), Support Vector Machine (SVM), Artificial Neural Network (ANN) and Linear Regression (LR). The figure has an x-axis limited to 13 for representation. The abbreviations used in the figure are expanded in Supplementary file S1.