# Peer review of "Modelling Point Mass Balance for the Glaciers of Central European Alps using Machine Learning Techniques"

_EGUsphere, 2022_

## Referee Comment (RC1)

**Review of "Modelling the Point Mass Balance for the Glaciers of Central European Alps using Machine Learning Techniques" by Anilkumar et al.**

EGUsphere

**1 General comments**

Anilkumar and colleagues present a study in which they use multiple machine learning methods to model point glacier mass balance for glaciers in the European Alps. This study is timely, thorough and provides new interesting insights on the use of machine learning to model glacier mass balance. As the authors explain, it provides the next logical step to the mass balance machine learning modelling literature: tackling point mass balance and using other methods than neural networks. Moreover, a recent study in the machine learning community demonstrated that for tabular data (like the one is normally used for glacier mass balance), tree-based models still outperform neural networks for various sizes of datasets[1]. This study corroborates those findings and provides new clues on the best way to model glacier mass balance using machine learning. For all this, I believe it represents a valuable contribution to the community.

Without taking away any of the merits of the study, I still believe there are multiple aspects of the study that could be improved in order to make the results more solid and easily understandable. For this, I will address some of them in the general comments (GC) section, and then I will provide detailed comments for different aspects throughout the text.

**1.1 GC1: Separation of train, validation and test datasets**

In my oppinion, the main weakness of the study right now is the way the cross-validation has been performed. For what I understood, the authors chose a classic 70% training - 30% test split. But some confusion remains regarding the wording, since the authors sometimes say that they use the test dataset for hyperparameter selection. I have two main issues with this:

**1) Have you used the test dataset for anything than just assessing the final performance of the model?** Hyperparameter selection should be done only with the validation dataset (i.e. using cross-validation in the training dataset). Using the test dataset for hyperparameter selection is considered a bad practice and will result in a clear model overfitting. Please confirm this and make the necessary changes if otherwise these necessary guidelines have not been followed.

**2) Why have you chosen a 3-fold cross-validation for hyperparameter selection?** This choice seems extremely arbitrary, and despite being a rather small number (the rule of

thumb is more like 5 to 10), it is particularly bothering because it probably implies that the folds have been randomly selected. This means that there is most likely a lot of leaked information in the train/validation folds, since it is quite likely that there are point mass balance data for a same glacier both in the train and validation folds, even for the same years. This information leakage makes the machine learning methods overfit, and could explain the reason why the authors have detected some potential overfitting.

When working with spatiotemporal data, it is essential to respect the spatiotemporal structures in the data (see Roberts et al., 2017[2] for a detailed explanation). This means, that folds should be designed in a way that they correctly separate the spatiotemporal instances that one is trying to model. First, the authors should determine if they aim at simulating point mass balance for unseen glaciers, unseen years, or both at the same time. Once this has been clarified, different strategies should be applied in choosing the folds, namely Leave-One-Group-Out, in order to ensure that there is no overlap in information between train and validation folds. This implies using cross-validation techniques such as Leave-One-Glacier-Out (or multiple glaciers), or Leave-One-Year-Out (or some years). A combination of both can also be used, which is probably what the authors want here. This is clearly explained in Roberts et al. (2017), and it was implemented for glacier-wide mass balance in Bolibar et al. (2020)[3].

I would ask the authors the revise their cross-validation methodology, and to try to design a strategy and clearly presented in a way that it avoids information leakage between train and validation folds. This separation strategy should also be applied to the test dataset, to avoid any overlap in terms of glaciers and years between train and test.

I have seen the authors have chosen to normalize input data between 0 and 1. Have you tried using other types of normalization such as the StandardScaler from Scikit-learn (i.e. substracting the mean and scaling to unit variance)?

Another aspect that would improve the intepretation of the results would be to understand how the errors relate to the target data. Right now, MSE are given for each model in mm w.e./yr. Could you please add a new figure with a histogram of the distribution of the point mass balance data from FoG? This would help understand what is the range of mass balance values and how those relate to the reported errors of the ML models. Having errors of 750 mm w.e./yr is not the same for a region with average MB rates of 100 mm w.e./yr than for regions with MB rates over a meter.

**1.2 GC2: Design of the variable training dataset experiment**

This is an aspect I particularly appreciated about this study. Such an experiment is very interesting to researchers in the field, since it gives important clues on which machine learning method might be most suitable for each case. Nonetheless, if I understood correctly the experiment design, I think that keeping the 30% test dataset constant and changing the size of the training dataset is not the best way to do this.

I believe that instead the total size of the full dataset (i.e. train + validation) should be changed, in order to respect the 30-70% ratio between train and test. Otherwise, adding new data will produce a different result depending on the correlation between those data points and the ones in the test dataset. This is particularly true in the context of the current (lack of) block cross-validation (see GC1). Since the authors have not correctly separated glaciers and

years between the train, validation, and test datasets, this effects will be even more enhanced.

Changing this should be rather straightforward, and would provide more reliable results to this interest experiment.

**1.3 GC3: Use of climate data from ERA5-Land**

One aspect that is not clear in the manuscript is how the climate data from ERA5 is used in the machine learning models. Since the authors are modelling point mass balance on glaciers, which are located on highly complex terrain, ERA5 is know to not capture well complex topography due to its coarse spatial resolution. It is unclear if the raw information from ERA5 has been used or if any downscaling or preprocessing has been performed.

Have you performed any correction on air temperature and precipitation to adjust to the glacier's altitude? How do you distinguish the different points in a glacier? For small European glaciers all of them probably fall inside the same ERA5 grid cell. If you don't perform any correction to temperature, how can actually extract different climate information for each mass balance point? Please explain this in more detail

These elements will also determine how much you can interpret the feature importance from a physical point of view. It would be interesting to bear in mind the limitations of the input climate dataset when interpreting each one of the machine learning models.

**1.4 GC4: Lack of perspectives**

This study introduces new methods, but offers almost no perspectives on what is the reason of their success and which new possibilities are opened by these. I would appreciate adding a section in which these aspects are discussed, and where the authors suggest the next steps, the main potential future bottlenecks, and what are the greatest opportunities following this study. Applying this to even more different glaciological regions will be challenging, especially in terms of cross-validation and hyperparameter tuning. How would you face those problems? Is there enough data available to apply this at a global scale? Answering such questions could be very useful for the community.

**2 Specific comments**

- **Title** I believe the title would sound better as "Modelling Point Mass Balance for the Glaciers of Central European Alps using Machine Learning Techniques".

- **L14-15** I'd rather present the RMSE (or MSE) in the abstract than the $r^2$, since it provides more information.

- **L35-36** I would also add the great number of parameters to calibrate.

- **L38-39** I would also point out the fact that for simulations over large temporal periods, temperature-index models (i.e. degree-day factors) are prone to be oversensitive to climatic changes[4].

- **L57** For me the sentence would read better as "and a nonlinear neural network..."

- **L59** This sentence is confusing. Artificial neural networks ARE machine learning models. I would reformulate, as you do in the abstract, to "have used the full diversity of different types of machine learning methods"

- **L68** Why use a linear regression example after mentioning NNs?

- **L98** I wouldn't call this training labels. This is a jargon more related to classification problems. I would just call them target data or reference data.

- **L100** Same with "labels".

- **L101** Regarding the parameters: that's the case for the NN only, right? Tree-based models don't really have parameters, mostly just hyperparameters to be tuned. Make sure that you really mean parameters and not hyperparameters.

- **L104** This would read better as "is a decision (regression or classification) tree".

- **L106** This would read better as "To illustrate this".

- **L122** The subject of the sentence is missing (i.e. "a neural network").

- **L125-126** "Nonlinearity" should be "nonlinearities".

- **L129** Same with "labelled data" and "labels".

- **L134** Why only annual mass balance observations and not seasonal? This is something that surprised me quite a lot, since dividing mass balance into accumulation and ablation season can definitely help to better calibrate melt vs accumulation features.

- **L143-144** Following GC3, please develop these aspects to make them clearer.

- **L146** As per GC1, please explain this better and make the corresponding changes.

- **L150** It's not the parameters which are tuned (e.g. the NNs weights), it's the hyperparameters. It's important not to confuse both.

- **L152** This is in fact cross-validation. So instead of just using a subset for validation you divide into folds.

- **L154** Following GC2, please better explain this and make the necessary changes.

- **L156** Do you mean the validation score? The test score can only be accessed once at the end, once you have selected the hyperparameters. Using the test dataset for selecting hyperparameters is a bad practice.

- **L157-158** What is the advantage of doing this? An advantage of the RMSE is that it keeps the units and it is therefore interpretable in terms of magnitude.

- **L168** Did you try any other activation functions? ReLu is known for vanishing. Did you try other improved activation functions such as Leaky ReLu or softplus?

- **L202** RMSE: Once the acronym has been introduced, you should use it to keep things brief.

- **L209** Please see GC2.

- **L284** Why are all the test performances given in these sections higher than the ones reported in the figures? Could you please explain and fix if this is an issue?

- **L320-321** This sentence is not clear, and seems somewhat contradictory. Could you please elaborate?

- **L324** I think you mean hyperparameters here.

- **L325** Please, revise the concepts of parameters and hyperparameters and make sure to use them correctly throughout the text.

- **L336** Tree-based models also provide a feature importance analysis in order to understand the most important input features. Did you compare the outpout of these with the permutation analyses? Are the results similar?

- **L352-353** This sentence is not clear, and seems somewhat contradictory. If you say that albedo is very important for the ablation season, why do you then say that is not important? Surface albedo is critical in summer, since the transition between snow, firn and ice drives important nonlinear spatial responses in terms of melt patters and the total annual mass balance.

- **L362-364** This is a very interesting finding and in line with recent studies from the machine learning community regarding ML for tabular data (cite[1])

- **L376-377** "We suggest the use of kernel-based model in such situations": This sentence appears out of the blue and it is not clear. Please merge with the following one to make your point clear.

- **Table 1** Please clearly separate each line in order to make it easy to see which hyperparameters are related to which model.

- **Figure 1** Here you should mention the validation dataset and call it 3-fold cross-validation, not validation.

- **Figure 2** Why are the errors reported here substantially lower than the ones reported in the text? Are you talking about different errors? Also, please report the units of the error in the vertical axis.

- **Figure 4** Please use target or reference data instead of "labelled". Why are the errors in here different than in Fig. 2?

- **Figure 5-7** These figures are not that interesting by themselves. I would either merge them in a single figure or move them to a supplementary material.

- **Figure 8** Instead of giving the abbreviations in the supplementary material, I think it would be better for the reader to have them in the legend. This should take that much space and it would increase readability.

**References**

1. Grinsztajn, L., Oyallon, E. & Varoquaux, G. *Why do tree-based models still outperform deep learning on tabular data?* 2022. https://arxiv.org/abs/2207.08815.

2. Roberts, D. R. *et al.* Cross-validation strategies for data with temporal, spatial, hierarchical, or phylogenetic structure. en. *Ecography* **40,** 913–929. ISSN: 09067590. http://doi.wiley.com/10.1111/ecog.02881 (2019) (Aug. 2017).

3. Bolibar, J. *et al.* Deep learning applied to glacier evolution modelling. en. *The Cryosphere* **14,** 565–584. ISSN: 1994-0424. https://www.the-cryosphere.net/14/565/2020/ (2020) (Feb. 2020).

4. Ismail, M. F., Bogacki, W., Disse, M., Schäfer, M. & Kirschbauer, L. Estimating degree-day factors based on energy flux components. *The Cryosphere Discussions* **2022,** 1–40. https://tc.copernicus.org/preprints/tc-2022-64/ (2022).

---

## Referee Comment (RC2)

**Review of: "Modelling the Point Mass Balance for the Glaciers of Central European Alps using Machine Learning Techniques" by Anilkumar et al.**

**Summary**

In this paper, the capabilities of different machine learning (ML) models in predicting point glacier mass balance are explored. The used data is composed of monthly meteorological data from ERA5-Land together with direct mass balance measurements in Central Europe from the Fluctuations of Glaciers database. The study is an important next step to explore which ML models are most suitable for applications of mass balance estimates. Further, they assess the data required for the different models and the importance of each meteorological variable. Both are very interesting and important questions for the potential future use of ML models in this field, also in light of increasing data availability in the future.
The study is well designed, but I think parts could be improved to make the results more solid and the manuscript easier to follow by the readers. I have divided my proposed changes into General Comments and specific/Line by Line comments.

**General Comments**

- **GC1**: I think it would be good to give more information on the values of the used mass balance observations (e.g. How is the distribution? Are they located in the yearly ablation regions of glaciers or also some in the accumulation regions?). Also, why are only annual mass balance observations used and no seasonal ones? This probably could improve the analysis of Feature Importance performed separately for accumulation and ablation months.

- **GC2**: If just the raw ERA5-Land data is used as input it is probably hard to asses feature importance due to the very complex topography which is poorly represented. In general, how did you deal with the downscaling of the meteorological data to the glacier location? In particular, how do you deal with the height difference between the ERA5-Land grid point and the glacier elevation or how do you deal with poorly resolved precipitation? (Could this be an explanation for why you could not find the expected importance of precipitation during the accumulation months?)

- **GC3**: Results sections 3.2, 3.3, 3.4, 3.5, 3.6: The last sentences of the first paragraph are not needed and could be incorporated at the end of the sentences where relevant things are discussed, e.g. '(Fig. 3).' at the end of the sentence, like is done in L307. This makes it easier for the reader to check your described findings by themselves in the plots. In the second paragraph, you can point to that this information is available in the supplementary in more detail.

- **GC4**: To make it easier for the reader to interpret the Figures you could include subfigure tags (e.g. (a), (b), (c), …) and describe in the Caption more precisely what is shown in each subfigure. Also, increase the font size where needed.

- **GC5**: You should use the same units in the text and figures, e.g. in the text L241 it says RMSE value of 1.071 mwe, but in Figure 3 the y-axis shows 1071 (with no unit given).

**Specific comments**

- **L190**: define which months are accumulation months and which months are ablation months, should be done earlier in the manuscript (is only defined in L335)

- **L204**: How is stabilizing the training metrics defined? We can not see this from Figure 2, maybe include a similar subplot as the right one for training performance.

- **L207**: Also here, how is stabilizing defined? 'This suggests that all models have successfully fit the data.': Doesn't it only shows that the results do not get better if we give the models more data than 50%, and it tells us nothing about how successful the fit is?

- **L209**: also here a plot suggested under L204 would be helpful to see the explained increase in training MAE

- **L217**: How do you see this? (Smaller box in Figure 2 left?)

- **L240**: Instead of 'This is depicted in Fig 5.' just right '(Fig. 5).' at the end of the sentence

- **L247**: define somewhere in the manuscript what are 'ablation meteorological variables'

- **L261**: is 'cost' the same as 'penalty'? If so you should be consistent and use one or the other throughout the manuscript.

- **L304**: How do you conclude this ranking? From Figure 3 and Figure 4, it looks like RF and SVM are closer than SVM and NN.

- **L326**: To which graphs are you linking here? Maybe include the figure number.

- **L348**: Probably you could not find the expected importance of precipitation because it is poorly resolved in the climate input data (see GC2).

- **Table 1**: Why is 'Number of trees' listed two times?

- **Figure 2**: See GC4. In the caption also explain which quantiles are shown in the box plot on the left. And explain how the two plots are connected (are the yellow boxes on the left representing the quantiles of the lines in the right plot?) Add the unit to the y-axis. Currently wrong caption: 'Training and testing RMSE (in mm we) and r values for varying the size of the training dataset for each of the models:' but only shown is MAE.

- **Figure 3**: See GC4. In caption: e.g. how are training and testing data split in this plot, 70%/30% or different, include (a), (b), (c) and (d) and explain also in the caption which performance measure is shown in which subplot. Add units to the y-axis where needed.

- **Figure 4**: See GC4. Maybe you can include the information of Figure 3 into this figure and delete Figure 3 (e.g. "RMSE: 0.95/1.08 mwe" and include a legend at the empty subplot space lower right with "RMSE: Training/Testing"). For the y-equations don't write y=0.744x + (-338.433) instead write y = 0.744x – 338.433. Is the high precision of numbers with three decimals meaningful for the RMA regression?

- **Figure 5**: hard to distinguish in the legend what are the solid lines and what are the dash-dotted lines. In the caption mention which test score is shown and explain briefly what the negative scaled RMSE is.

- **Figure 6**: In the caption mention which test score is shown and explain briefly what the negative scaled RMSE is.

- **Figure 7**: increase the font size, In Caption mention which test score is shown. Also include the test score name in the y-axis (currently only 'Test score').

- Maybe you could combine Figures 5, 6 and 7 into one Figure.

- **Figure 8**: increase the font size. Because the x-axis is limited to 13 maybe add the numbers in the plot for features which go beyond this limit. Maybe include the abbreviations of meteorological variables in the caption or the text, so you can understand the plot without having a look in the supplementary. And you can also use the abbreviations in the result sections.

- **Supplementary S1**:
    - general: give more meaningful names to the individual sheets
    - sheet3: no explanation of what is shown on this sheet, include references in the text or delete this sheet

---

## Author Comment (AC1)

**Modelling the Point Mass Balance for the Glaciers of Central European Alps using Machine Learning Techniques**

Ritu Anilkumar, Rishikesh Bharti, Dibyajyoti Chutia, Shiv Prasad Aggarwal
**Correspondance:** ritu.anilkumar@nesac.gov.in

We are grateful for the reviewer's detailed and insightful comments on manuscript number **egusphere-2022-1076**: 'Modelling the Point Mass Balance for the Glaciers of Central European Alps using Machine Learning Techniques'. The point-by-point response to the comments is provided below. The comments by the reviewer are quoted in *a black font color and italicised font style*. The author's response is in blue font color and normal font style. Text quoted from the revised manuscript is ***blue font color and bold italicized font style***.

**1 General Comments:**

*Anilkumar and colleagues present a study in which they use multiple machine learning methods to model point glacier mass balance for glaciers in the European Alps. This study is timely, thorough and provides new interesting insights on the use of machine learning to model glacier mass balance. As the authors explain, it provides the next logical step to the mass balance machine learning modelling literature: tackling point mass balance and using other methods than neural networks. Moreover, a recent study in the machine learning community demonstrated that for tabular data (like the one is normally used for glacier mass balance), tree-based models still outperform neural networks for various sizes of datasets. This study corroborates those findings and provides new clues on the best way to model glacier mass balance using machine learning. For all this, I believe it represents a valuable contribution to the community.*

*Without taking away any of the merits of the study, I still believe there are multiple aspects of the study that could be improved in order to make the results more solid and easily understandable. For this, I will address some of them in the general comments (GC) section, and then I will provide detailed comments for different aspects throughout the text.*

**1.1 COMMENT GC1: Separation of train, validation and test datasets**

*In my oppinion, the main weakness of the study right now is the way the cross-validation has been performed. For what I understood, the authors chose a classic 70% training - 30% test split. But some confusion remains regarding the wording, since the authors sometimes say that they use the test dataset for hyperparameter selection. I have two main issues with this:*

*1) Have you used the test dataset for anything than just assessing the final performance of the model? Hyperparameter selection should be done only with the validation dataset (i.e. using cross-validation in the training dataset). Using the test dataset for hyperparameter selection is considered a bad practice and will result in a clear model overfitting. Please confirm this and make the necessary changes if otherwise these necessary guidelines have not been followed.*

Thank you for pointing out the lack of clarity in the manuscript pertaining to the validation and testing dataset. We have used a 70%-30% split for training and testing. Here, 30% is reserved for assessing the model performance only. It has not been used for hyperparameter selection. For hyperparameter selection, we use a 3 fold cross validation using the 70% training split. The modifications will be included in the revised manuscript. Line 146 in the original manuscript 'We split the dataset into training and testing samples to be utilised by the model' is revised to: **We have split the dataset using a random split where 70% of the total dataset is used for training the model and 30% is used for testing the model performance. The training split is used in a 3 fold cross validation process for tuning the hyperparameters as described further in Section 2.3**

*2) Why have you chosen a 3-fold cross-validation for hyperparameter selection? This choice seems extremely arbitrary, and despite being a rather small number (the rule of thumb is more like 5 to 10), it is particularly bothering because it probably implies that the folds have been randomly selected. This means that there is most likely a lot of leaked information in the train/validation folds, since it is quite likely that there are point mass balance data for a same glacier both in the train and validation folds, even for the same years. This information leakage makes the machine learning methods overfit, and could explain the reason why the authors have detected some potential overfitting.*

*When working with spatiotemporal data, it is essential to respect the spatiotemporal structures in the data (see Roberts et al., 2017 for a detailed explanation). This means, that folds should be designed in a way that they correctly separate the spatiotemporal instances that one is trying to model. First, the authors should determine if they aim at simulating point mass balance for unseen glaciers, unseen years, or both at the same time. Once this has been clarified, different strategies should be applied in choosing the folds, namely Leave-One-GroupOut, in order to ensure that there is no overlap in information between train and validation folds. This implies using cross-validation techniques such as Leave-One-Glacier-Out (or multiple glaciers), or Leave-One-Year-Out (or some years). A combination of both can also be used, which is probably what the authors want here. This is clearly explained in Roberts et al. (2017), and it was implemented for glacier-wide mass balance in Bolibar et al. (2020).*

*I would ask the authors the revise their cross-validation methodology, and to try to design a strategy and clearly presented in a way that it avoids information leakage between train and validation folds. This separation strategy should also be applied to the test dataset, to avoid any overlap in terms of glaciers and years between train and test.*

Thank you for your suggestion. We have maintained a 70-30 ratio for the training and testing datasets. The reason we went for a 3 fold cross validation is to maintain the data folds in a manner that best replicated the ratio used for the overall training and testing. Among the 3

folds, for a given iteration, 2 folds are used for training and 1 for validation. This 2:1 ratio of training and validation is the closest representation of the 70:30 split of the training and test datasets.

Regarding the cross validation and testing data split methodology, we accept the point made by the reviewer that spatially and temporally structured datasets would benefit from a manually designed blocking strategy such as the Leave One Glacier Out and Leave One Year Out strategy as depicted in Bolibar et al 2020. Acknowledging the merits of this technique in validation and testing split generation, we would like to bring to the attention of the reviewer that through this study, we aim to perform a comparative assessment of a number of machine learning models available using present day data driven techniques and explain the feature importance associated with a machine learning modelling of the glacier mass balance. As the testing and validation splits will result in similar effects in all the models, performing an additional exercise in blocking strategies of the data split might dilute the information we wish to convey. However, for cases where a single model is to be used to estimate glacier mass balance, we accept that the Leave One Glacier Out and Leave One Year Out techniques are vital. We propose including this in subsection 4.4 Relevance to future studies under Discussions (see response to GC4) and incorporating a section in the supplementary material with a comparison of the effect of the blocking strategy and the random split performance.

*I have seen the authors have chosen to normalize input data between 0 and 1. Have you tried using other types of normalization such as the StandardScaler from Scikit-learn (i.e. substracting the mean and scaling to unit variance)?*

Thank you for this point. We will perform this exercise and include this as an additional exercise in the supplementary material demonstrating the role of the min-max 0 to 1 scaling compared to the StandardScaler that performs a mean subtract and scaling by standard deviation.

*Another aspect that would improve the intepretation of the results would be to understand*

*how the errors relate to the target data. Right now, MSE are given for each model in mm w.e./yr. Could you please add a new figure with a histogram of the distribution of the point mass balance data from FoG? This would help understand what is the range of mass balance values and how those relate to the reported errors of the ML models. Having errors of 750 mm w.e./yr is not the same for a region with average MB rates of 100 mm w.e./yr than for regions with MB rates over a meter.*

We are grateful to the reviewer for bringing this to our attention. We can generate the figure of the glacier mass balance measurements as a histogram to provide a complete representation of the mass balance and the errors to the readers. Further, we can include an additional error metric normalized root mean square error (nRMSE) which depends upon the root mean square error (RMSE) and the standard deviation of mass balance measurements ($\sigma_{MB}$) defined as:

$$nRMSE = \frac{RMSE}{\sigma_{MB}} \tag{1}$$

to represent the errors including the context of the natural variation in the target data. This will be included in the revised manuscript.

**1.2    COMMENT GC2: Design of the variable training dataset experiment**

*This is an aspect I particularly appreciated about this study. Such an experiment is very interesting to researchers in the field, since it gives important clues on which machine learning method might be most suitable for each case. Nonetheless, if I understood correctly the experiment design, I think that keeping the 30% test dataset constant and changing the size of the training dataset is not the best way to do this.*

*I believe that instead the total size of the full dataset (i.e. train + validation) should be changed, in order to respect the 30-70% ratio between train and test. Otherwise, adding new data will produce a different result depending on the correlation between those data points and the ones in the test dataset. This is particularly true in the context of the current (lack of)*

*block cross-validation (see GC1). Since the authors have not correctly separated glaciers and years between the train, validation, and test datasets, this effects will be even more enhanced. Changing this should be rather straightforward, and would provide more reliable results to this interest experiment.*

We thank the reviewer for pointing out the ambiguity in the text explaining how the training and testing split was undertaken. In fact, we have maintained the ratio of a 70:30 split consistently for varying dataset sizes to ensure a complete separation between the training and testing samples. In order to explain this, we modify line 180-182 in the original submission as follows:

**"To understand the effect of data availability on the model performance, we perform an experiment on varying the training sizes. We split the original dataset into subsets of iteratively increasing sizes. We partition each subset into training and testing partitions using a 70:30 ratio. For each subset, we train all the models using the training partition and computed the evaluation metrics over the testing partition. "**

**1.3    COMMENT GC3: Use of climate data from ERA5-Land**

*One aspect that is not clear in the manuscript is how the climate data from ERA5 is used in the machine learning models. Since the authors are modelling point mass balance on glaciers, which are located on highly complex terrain, ERA5 is know to not capture well complex topography due to its coarse spatial resolution. It is unclear if the raw information from ERA5 has been used or if any downscaling or preprocessing has been performed.*

*Have you performed any correction on air temperature and precipitation to adjust to the glacier's altitude? How do you distinguish the different points in a glacier? For small European glaciers all of them probably fall inside the same ERA5 grid cell. If you don't perform any correction to temperature, how can actually extract different climate information*

*for each mass balance point? Please explain this in more detail.*

*These elements will also determine how much you can interpret the feature importance from a physical point of view. It would be interesting to bear in mind the limitations of the input climate dataset when interpreting each one of the machine learning models.*

150 We thank the reviewer for pointing out the the lack of clarity in the text explaining how the climate datasets have been used. We agree ERA5 is known to be erroneous in complex terrains due to its course spatial resolution (31 km/pixel). We have used the ERA5-Land product which is generated by integrating the ECMWF land surface model driven by the downscaled meteorological forcing from the ERA5 climate reanalysis dataset. This included

155 an altitude correction and is described further in Muñoz-Sabater et al 2021. The final product we have used has a spatial resolution of 9 km/pixel.

We acknowledge that this resolution of 9km/pixel is also large as we are using point glacier mass balance measurements. Applying a scaling factor can be straightforward in case of features such as temperature. However, choosing appropriate scaling factors for other me-

160 teorological variables (e.g sensible and latent heat fluxes, albedo) is not intuitive. While we accept that the effects of the larger scale of the input variable will persist in the model, we note that the effects will be consistent across all the models. Thus the effect of the input variable scale is represented by the uncertainty of all models. This will be described further in the subsection 4.1 Comparison of Model Performance and Associated Errors under

165 Discussions.

**1.4 COMMENT GC4: Lack of perspectives**

*This study introduces new methods, but offers almost no perspectives on what is the reason of their success and which new possibilities are opened by these. I would appreciate adding a section in which these aspects are discussed, and where the authors suggest the next steps,*

170 *the main potential future bottlenecks, and what are the greatest opportunities following this*

*study. Applying this to even more different glaciological regions will be challenging, especially in terms of cross-validation and hyperparameter tuning. How would you face those problems? Is there enough data available to apply this at a global scale? Answering such questions could be very useful for the community.*

Thank you for bringing this point to our notice. As suggested, we will incorporate a subsection under Discussions titled '4.4 Relevance to future studies' in the revised manuscript which will include the following points:

- The continued importance of tree based methods for tabular data structures keeping in mind the suggestions provided in Grinsztajn et al 2022

- Guidelines on the use of machine learning techniques for future studies

- Extension of the study to other datasets available

- Extension of the study to other data sparse RGI regions with emphasis on the role of transfer learning.

- Reducing uncertainties of the models by downscaling of input variables.

- Understanding the role of feature importance for different glaciated regions and the importance of local, regional and global inputs.

**2 Specific Comments:**

1 **Reviewer Comment:** *Title I believe the title would sound better as "Modelling Point Mass Balance for the Glaciers of Central European Alps using Machine Learning Techniques".*

**Author Response:** Thank you. We agree the title Modelling Point Mass Balance for the Glaciers of Central European Alps using Machine Learning Techniques sounds better. We will incorporate this change in the updated manuscript.

**2 Reviewer Comment:** *L14-15 I'd rather present the RMSE (or MSE) in the abstract than the $r^2$, since it provides more information*

**Author Response:** Thank you. We agree with the comment and will include the RMSE in the abstract as well.

**3 Reviewer Comment:** *L35-36 I would also add the great number of parameters to calibrate.*

**Author Response:** Thank you. We will modify the lines in the revised manuscript as follows: ***However, the substantial requirement for ground data to force the model, the sizeable number of parameters to calibrate and the computational complexity associated with running the model make it cumbersome to use for large areas***

**4 Reviewer Comment:** *L38-39 I would also point out the fact that for simulations over large temporal periods, temperature-index models (i.e. degree-day factors) are prone to be oversensitive to climatic changes*[4]

**Author Response:** Thank you for directing us to this study. We agree, the variations in DDFs spatially and temporally are significant. We will include the following lines in the text: ***However, using only temperature and precipitation as inputs can lead to oversimplification. Further, the degree day factors (DDF) considered in temperature index models are often invariant. But studies such as Gabbi et al 2014, Matthews and Hodgins 2016, Ismail et al 2022 have observed a decreasing trend in DDF, particularly in higher elevations. Ismail et al 2022 also report the sensitivity of the DDF under the influence of the changing climate, particularly to to solar radiation and albedo.***

**5 Reviewer Comment:** *L57 For me the sentence would read better as "and a nonlinear neural network..."*

**Author Response:** We agree. The sentence is fixed to read as: ***Bolibar et al. (2020) used a least absolute shrinkage and selection operator (LASSO) regression, a linear model, and a nonlinear neural network model to simulate glacier mass balance.***

**6 Reviewer Comment:** *L59 This sentence is confusing. Artificial neural networks ARE machine learning models. I would reformulate, as you do in the abstract, to "have used the full diversity of different types of machine learning methods"*

**Author Response:** Thank you for bringing this to our notice. We have modified lines 57-62 in the updated manuscript as follows: ***Steiner et al. (2005); Vincent et al. (2018); Bolibar et al. (2020, 2022) are some of the few studies reporting consistently better performance of non-linear models over linear models. These studies have largely used neural networks. However, a gamut of ML techniques such as ensemble-based and kernel-based techniques exist which have largely been under-utilized for the purpose of modelling glacier mass balance.***

**7 Reviewer Comment:** *L68 Why use a linear regression example after mentioning NNs?*

**Author Response:** Thank you for pointing out this oversight. We meant to represent the inputs used in data driven models, not neural networks specifically here. We have corrected line 67 in the original manuscript to: ***Existing data-driven models typically use a subset of topographic and meteorological variables.***

**8 Reviewer Comment:** *L98 I wouldn't call this training labels. This is a jargon more related to classification problems. I would just call them target data or reference data.*

**Author Response:** Thank you. We agree. All instances of training labels will be replaced by target data.

**9 Reviewer Comment:** *L100 Same with "labels"*

**Author Response:** Thank you. This is fixed.

**10 Reviewer Comment:** *L101 Regarding the parameters: that's the case for the NN only, right? Tree-based models don't really have parameters, mostly just hyperparameters to be tuned. Make sure that you really mean parameters and not hyperparameters.*

**Author Response:** Thank you for pointing out the lack of consistency in this usage. We have fixed all occurrences to reflect the correct term.

**11 Reviewer Comment:** *L104 This would read better as "is a decision (regression or classification) tree".*

**Author Response:** Thank you. This modification has been incorporated.

**12 Reviewer Comment:** *L106 This would read better as "To illustrate this".*

**Author Response:** Thank you. This modification has been incorporated.

**13 Reviewer Comment:** *L122 The subject of the sentence is missing (i.e. "a neural network").*

**Author Response:** Thank you. We have corrected this sentence as follows: **Hornik (1991) showed that neural networks with as few as a single hidden layer with a sufficiently large number of neurons, when used with a non-constant unbounded activation function, can function as universal function approximators.**

**14 Reviewer Comment:** *L125-126 "Nonlinearity" should be "nonlinearities".*

**Author Response:** Thank you. This modification has been incorporated.

**15 Reviewer Comment:** *L129 Same with "labelled data" and "labels".*

**Author Response:** Thank you. This modification has been incorporated.

**16 Reviewer Comment:** *L134 Why only annual mass balance observations and not seasonal? This is something that surprised me quite a lot, since dividing mass balance into accumulation and ablation season can definitely help to better calibrate melt vs accumulation features.*

**Author Response:** Thank you for this suggestion. We considered annual mass balance observations purely due to availability of data. The database of point glacier mass balance observations contain separate entries for annual mass balance observations, summer and winter mass balance. For example, we have 9595 points using annual mass balance observations after 1950. For accumulation season, 3281 points are available and for ablation season only 1783 points are available. While we do agree with the reviewer that separation of mass balance can help bring out the features associated with the accumulation and ablation, a combined measurement of summer, winter and annual point mass balance for the same location was not available using this database.

**17 Reviewer Comment:** *L143-144 Following GC3, please develop these aspects to make them clearer.*

**Author Response:** Thank you. This point is addressed in the response to GC3.

**18 Reviewer Comment:** *L146 As per GC1, please explain this better and make the corresponding changes.*

**Author Response:** Thank you. We have incorporated the suggestion as described in the response of GC1.

**19 Reviewer Comment:** *L150 It's not the parameters which are tuned (e.g. the NNs weights), it's the hyperparameters. It's important not to confuse both.*

**Author Response:** Thank you for bringing this to our notice. We will correct all instances of the misuses of terms hyperparameters and parameters

**20 Reviewer Comment:** *L152 This is in fact cross-validation. So instead of just using*

*a subset for validation you divide into folds.*

**Author Response:** Yes, we have modified the lines 151-156 (Rather than using...optimal hyperparameters are selected) for improved clarity as follows: ***We have considered a hyperparameter grid with all combinations of values that each hyperparameter can take (see Table 1). Rather than using a fixed ratio subset for validation as was the case with the testing, we divided the training data subset into three equal folds. Two folds are randomly selected as the training set and the third fold is used for validation. The validation score is noted and the process is then repeated for the other fold combinations. The mean validation score for each hyperparameter setting obtained from the grid is used for selection of the optimal hyperparameters.***

**21 Reviewer Comment:** *L154 Following GC2, please better explain this and make the necessary changes.*

**Author Response:** Thank you. The changes have been incorporated as specified in the previous comment response.

**22 Reviewer Comment:** *L156 Do you mean the validation score? The test score can only be accessed once at the end, once you have selected the hyperparameters. Using the test dataset for selecting hyperparameters is a bad practice.*

**Author Response:** Thank you for pointing this out. Yes, we did intent to write validation score. We have corrected it at all occurances of this error.

**23 Reviewer Comment:** *L157-158 What is the advantage of doing this? An advantage of the RMSE is that it keeps the units and it is therefore interpretable in terms of magnitude.*

**Author Response:** In the k fold validation technique, we have considered all permutations of folds as training and testing subsets. For example, for our case where 3 folds were used, First fold 1 and 2 were used for training and 3 for assessment (validation). Then 2 and 3 for training and 1 for assessment. Finally 1 and 3 were used for training and 2 for assessment. There are thus 3 assessment values that we obtain. The mean score of this is the final validation score which is used to represent each hyperparameter setting. While the rescaled RMSE provides an estimate of errors that is useful at the time of reporting accuracies, for a comparative analysis, the relative values are sufficient. Hence the scaling back to original units was not undertaken. Further, we used negative of the RMSE purely for intuition in assigning ranks to the hyperparameter combination setting. Settings with higher RMSEs perform poorly. Thus settings with higher negative RMSE perform well and can be ranked better.

**24 Reviewer Comment:** *L168 Did you try any other activation functions? ReLu is known for vanishing. Did you try other improved activation functions such as Leaky ReLu or softplus?*

**Author Response:** Thank you for bringing this up. We did try alternate runs with Parametrized Leaky ReLU on PyTorch. We did not include it in the final version of the manuscript for brevity. We can include the sample runs as well as the code for the same in the Supplementary material.

**25 Reviewer Comment:** *L202 RMSE: Once the acronym has been introduced, you should use it to keep things brief.*

**Author Response:** Thank you. This is fixed.

**26 Reviewer Comment:** *L209 Please see GC2*

**Author Response:** Thank you. We have incorporated the changes described in response to GC2 at line 180.

**27 Reviewer Comment:** *L284 Why are all the test performances given in these sections*

*higher than the ones reported in the figures? Could you please explain and fix if this is an issue?*

**Author Response:** These errors are the same as those depicted in Figures 3 and 4 of the original submission. It is different from Figure 2 as the mean absolute error is reported in Figure 2 not the Root Mean Squared Error. We will include the mean absolute errors in addition to the RMSE in the revised manuscript text for all models.

**28 Reviewer Comment:** *L320-321 This sentence is not clear, and seems somewhat contradictory. Could you please elaborate?*

**Author Response:** Thank you for bringing this to our notice. For clarity, we will rewrite the lines 320-322 as follows: ***The testing performance improves on increasing the number of training samples. We observe that for larger number of data points, marginal improvement is observed upon increasing the number of samples further. The reduction in rate of improvement for all models suggest that all models have been successfully trained. However, the marginal improvements observed suggest a potential improvement in model performance is is possible when including more data samples.***

**29 Reviewer Comment:** *L324 I think you mean hyperparameters here.*

**Author Response:** Thank you for your comment. We mean parameters here as we are referring to the weights and not the hyperparameters such as number of layers, number of neurons or activation.

**30 Reviewer Comment:** *L325 Please, revise the concepts of parameters and hyperparameters and make sure to use them correctly throughout the text.*

**Author Response:** Thank you. Yes, here, we mean the weights associated with the network and hence parameters was used. Other erroneous misuse of the terms parameter and hyperparameters have been corrected in the revised version of the manuscript.

**31 Reviewer Comment:** *L336 Tree-based models also provide a feature importance analysis in order to understand the most important input features. Did you compare the outpout of these with the permutation analyses? Are the results similar?*

**Author Response:** Tree based models do provide a feature importance analysis. However, these use a mean decrease in impurity (RMSE, MSE for regression or gini for classification). Strobl et al 2007 report a skewed representation of features in such cases as a result of varying scales of the data and correlation between the input features. In our study, normalization is performed. Thus, the varying scales of data will not be a hindrance. However, correlation is observed between the input variables. This renders tree based importance metrics less accurate. This issue is resolved using permutation importance. Thus we selected permutation importance . An additional advantage of using permutation importance is to be able to use a model-agnostic explainability metric.

**32 Reviewer Comment:** *L352-353 This sentence is not clear, and seems somewhat contradictory. If you say that albedo is very important for the ablation season, why do you then say that is not important? Surface albedo is critical in summer, since the transition between snow, firn and ice drives important nonlinear spatial responses in terms of melt patters and the total annual mass balance.*

**Author Response:** Thank you for bringing this to our notice. We correct this to the following:   ***Albedo over snow-covered regions is higher than that of exposed ice or firn. At higher elevations and in summer months, we expect the lower values of albedo. Thus variations in albedo are are significance. The expected importance of the albedo is observed in the RF, GBR, NN and SVM model. LR models, in contrast, depict a very low importance of albedo for the accumulation months.***

**33 Reviewer Comment:** *L362-364 This is a very interesting finding and in line with*

*recent studies from the machine learning community regarding ML for tabular data (cite1)*

395     **Author Response:** Thank you for your comment. In line with GC4, we are including a subsection in the Discussions where we will describe the importance of tree based ensembled in working with tabular datasets.

**34 Reviewer Comment:** *L376-377 "We suggest the use of kernel-based model in such situations": This sentence appears out of the blue and it is not clear. Please merge with* 400     *the following one to make your point clear.*

**Author Response:** Thank you. Yes, we will delete this sentence as the next sentence explains this better.

**35 Reviewer Comment:** *Table 1 Please clearly separate each line in order to make it easy to see which hyperparameters are related to which model.*

405     **Author Response:** Thank you. We have fixed this. The table now appears as depicted in Table 1:

**36 Reviewer Comment:** *Figure 1 Here you should mention the validation dataset and call it 3-fold cross validation, not validation.*

**Author Response:** Thank you. We have fixed this. The figure is as depicted in 410     Figure **R1**:

**37 Reviewer Comment:** *Figure 2 Why are the errors reported here substantially lower than the ones reported in the text? Are you talking about different errors? Also, please report the units of the error in the vertical axis.*

**Author Response:** Thank you. Yes, here, we used the mean absolute error and the 415     errors reported in the text are the root mean squared error. For clarity, we will include mean absolute error in the text for each model.

Table 1: Grid of settings used for hyperparameter tuning of each of the models

| Machine learning model | Hyperparameter | Values |
|---|---|---|
| Random Forest | Number of trees | 10,20,50,100 |
| Gradient Boosted Regressor | Number of trees | 50,100,200 |
| | Subsampling | 0.7, 1.0 |
| | Maximum Depth | 3,5,10 |
| Support Vector Machine | Cost | 0.1, 1, 10, 20 |
| | Kernels | Sigmoid, Radial Basis Function, Polynomial |
| | Degree (polynomial kernel) | 2, 3, 4, 5 |
| Artificial Neural Network | Number of layers and nodes | **1:** 10, 50, 100, 200, 300, 400, 500, **2:** (100, 50), (200, 100), (400, 200), (200, 400) **3:** (400, 200, 100), (500, 200, 100), (200, 100, 50), (100, 50, 10), **4:** (200, 300, 400, 500), (300, 200, 100, 50), (200, 100, 50, 10) |

**38 Reviewer Comment:** *Figure 4 Please use target or reference data instead of "labelled". Why are the errors in here different than in Fig. 2?*

**Author Response:** Thank you, the labelling of figures will be corrected. The errors specified here are different from figure 2 because here, we depict the root mean squared error as opposed to mean absolute error in Figure 2.

**39 Reviewer Comment:** *Figure 5-7 These figures are not that interesting by themselves. I would either merge them in a single figure or move them to a supplementary material.*

**Author Response:** Thank you. We agree. We will merge them into a single figure for the final manuscript.

**40 Reviewer Comment:** *Figure 8 Instead of giving the abbreviations in the supplementary material, I think it would be better for the reader to have them in the legend. This should take that much space and it would increase readability.*

[Figure]

Figure **R1**: Flowchart of the methodology

**Author Response:** Thank you. We agree. To improve readability, we reformatted the image in the form of a RADAR plot with the labels on the right. The tentative figure is as depicted in Figure **R2**.

**References**

1. Muñoz-Sabater, J, Dutra, E, Agustí-Panareda, A, Albergel, C, Arduini, G, Balsamo, G, Boussetta, S, Choulga, M, Harrigan, S, Hersbach, H and Martens, B (2021) ERA5-Land: A state-of-the-art global reanalysis dataset for land applications. Earth System Science Data, 13(9), pp.4349-4383. doi: 10.5194/essd-13-4349-2021

2. Gabbi, J, Carenzo, M, Pellicciotti, F, Bauder, A and Funk, M (2014) A comparison of empirical and physically based glacier surface melt models for long-term simulations of

glacier response. J. Glaciol., 60(224), 1140–1154. doi: 10.3189/2014JoG14J011

3. Matthews, T., and Hodgkins, R. (2016). Interdecadal variability of degree-day factors on Vestari Hagafellsjökull (Langjökull, Iceland) and the importance of threshold air temperatures. Journal of Glaciology, 62(232), 310-322. doi:10.1017/jog.2016.21

4. Ismail, M. F. and Bogacki, W. and Disse, M. and Schäfer, M. and Kirschbauer, L. (2022). Estimating degree-day factors based on energy flux components. The Cryosphere Discussions 1-40, doi:10.5194/tc-2022-64

5. Grinsztajn, L., Oyallon, E., and Varoquaux, G. (2022). Why do tree-based models still outperform deep learning on tabular data?. arXiv preprint arXiv:2207.08815.

[Figure]

Figure **R2**: Radar plot depicting the percentage importance of all features summed over the accumulation and ablation season for the models: Random Forest, Gradient Boosted Regression, Support Vector Machine, Artificial Neural Network and Linear Regression. The radial axis represents the summed percentage importance and the angular axis represents the input features.

---

## Author Comment (AC2)

**Modelling the Point Mass Balance for the Glaciers of Central European Alps using Machine Learning Techniques**

Ritu Anilkumar, Rishikesh Bharti, Dibyajyoti Chutia, Shiv Prasad Aggarwal
**Correspondance:** ritu.anilkumar@nesac.gov.in

We are grateful for the reviewer's critical and insightful feedback on manuscript number **egusphere-2022-1076**: 'Modelling the Point Mass Balance for the Glaciers of Central European Alps using Machine Learning Techniques'. The point-by-point response to the comments is provided below. The comments by the reviewer are quoted in *a black font colour and italicised font style*. The author's response is in blue font colour and normal font style. Text quoted from the revised manuscript is *blue font colour and bold italicized font style*.

**1 Summary and General Comments:**

*In this paper, the capabilities of different machine learning (ML) models in predicting point glacier mass balance are explored. The used data is composed of monthly meteorological data from ERA5- Land together with direct mass balance measurements in Central Europe from the Fluctuations of Glaciers database. The study is an important next step to explore which ML models are most suitable for applications of mass balance estimates. Further, they assess the data required for the different models and the importance of each meteorological variable. Both are very interesting and important questions for the potential future use of ML models in this field, also in light of increasing data availability in the future.*
*The study is well designed, but I think parts could be improved to make the results more solid and the manuscript easier to follow by the readers. I have divided my proposed changes into General Comments and specific/Line by Line comments.*

**COMMENT GC1:** *I think it would be good to give more information on the values of the used mass balance observations (e.g. How is the distribution? Are they located in the yearly ablation regions of glaciers or also some in the accumulation regions?). Also, why are only annual mass balance observations used and no seasonal ones? This probably could improve the analysis of Feature Importance performed separately for accumulation and ablation months.*

Thank you for this suggestion. We agree that some more information pertaining to the point mass balance values considered will be beneficial for a complete interpretation of our findings. We will include a figure representing the distribution of the mass balance measurements with a histogram of measurement values to the manuscript.

Regarding the consideration of only annual mass balance observations as opposed to seasonal observations, our decision was purely a consequence of the availability of data. The database of point glacier mass balance observations contains separate entries for annual mass balance observations and seasonal mass balance. For example, we have 9595 points using annual mass balance observations after 1950. For accumulation season, 3281 points are available and for ablation season only 1783 points are available, all of which do not overlap with the accumulation season measurements. While we do agree with the reviewer that separation of mass balance can help bring out the features associated with the accumulation and ablation, a combined measurement of summer, winter and annual point mass balance for the same location was not available using the existing database.

**COMMENT GC2:** *If just the raw ERA5-Land data is used as input it is probably hard to asses feature importance due to the very complex topography which is poorly represented. In general, how did you deal with the downscaling of the meteorological data to the glacier location? In particular, how do you deal with the height difference between the ERA5-Land grid point and the glacier elevation or how do you deal with poorly resolved precipitation? (Could this be an explanation for why you could not find the expected importance of precipitation*

*during the accumulation months?)*

Thank you for your suggestion. We acknowledge that this resolution of 9km/pixel is a poor representation to model glacier mass balance measurements at point scale. Approaches such as using a scaling factor or lapse rates have been attempted by studies ( e.g. Radić et al 2014, Maussion et al 2019). However, these studies largely utilize precipitation and temperature as inputs, the scaling of which with elevation is fairly straightforward. Choosing appropriate scaling factors for other meteorological variables (e.g sensible and latent heat fluxes, albedo) is not intuitive. While we accept that the effects of the larger scale of the input variable will persist in the model, we would like to bring to notice that the effects will be consistent across all the models. Thus the effect of the input variable scale is represented by the uncertainty of all models. This will be described further in the subsection 4.1 Comparison of Model Performance and Associated Errors under Discussions.

***COMMENT GC3:*** *Results sections 3.2, 3.3, 3.4, 3.5, 3.6: The last sentences of the first paragraph are not needed and could be incorporated at the end of the sentences where relevant things are discussed, e.g. '(Fig. 3).' at the end of the sentence, like is done in L307. This makes it easier for the reader to check your described findings by themselves in the plots. In the second paragraph, you can point to that this information is available in the supplementary in more detail.*

We thank the reviewer for pointing this out. We accept incorporating this change will improve the readability of the manuscript. The necessary changes will be included in the revised manuscript

***COMMENT GC4:*** *To make it easier for the reader to interpret the Figures you could include subfigure tags (e.g. (a), (b), (c), ...) and describe in the Caption more precisely what is shown in each subfigure. Also, increase the font size where needed.*

Thank you for your suggestion. We will modify all the figures accordingly.

***COMMENT GC5:*** *You should use the same units in the text and figures, e.g. in the text*

*L241 it says RMSE value of 1.071 mwe, but in Figure 3 the y-axis shows 1071 (with no unit given).*

Thank you for pointing out this oversight. For the next revision, we will ensure that all units are specified and consistent throughout the manuscript.

**2 Specific Comments:**

1 **Reviewer Comment:** *L190: define which months are accumulation months and which months are ablation months, should be done earlier in the manuscript (is only defined in L335)*

**Author Response:** Thank you. We agree with the suggestion. We will modify lines 189-191 as follows:

**(b) Percentage importance associated with the accumulation months (November to March) and the ablation months (June-September) are summed and graphically represented for each model in Fig. 8.**

2 **Reviewer Comment:** *L204: How is stabilizing the training metrics defined? We can not see this from Figure 2, maybe include a similar subplot as the right one for training performance.*

**Author Response:** Thank you for bringing this to our notice. We will include the Figure **R1** as a subplot to Figure 2 of the manuscript for clarity.

3 **Reviewer Comment:** *L207: Also here, how is stabilizing defined? 'This suggests that all models have successfully fit the data.': Doesn't it only shows that the results do not get better if we give the models more data than 50%, and it tells us nothing about how successful the fit is?*

**Author Response:** Thank you for bringing this to our notice. The term stabilizing is defined as "no significant change in the metric." Here, we have considered the change

[Figure]

Figure **R1**

in performance greater than 50mm we to be a significant change in mean absolute error. Further, regarding the sentence 'This suggests that all models have successfully fit the data,' we agree with the reviewer. For clarity, we rewrite the lines 204 to 208 as follows ***The training metrics do not show significant change after 20-30% of the training dataset size for the LR, RF, GBR and SVM models and after 40% for the NN model. This illustrates the larger number of trainable parameters resulting in the requirement of larger datasets for artificial neural networks for training. The testing performance of each of the models do not show significant change for training dataset sizes larger than 50%. We observe that while a downward trend is evident with the addition of new data, the rate of improvement is slower.***

**4  Reviewer Comment:** *L209: also here a plot suggested under L204 would be helpful to see the explained increase in training MAE*

**Author Response:** Thank you for this suggestion. We will incorporate Figure **R1** as a subplot to Figure 2 of the manuscript.

**5  Reviewer Comment:** *L217: How do you see this? (Smaller box in Figure 2 left?)*

**Author Response:** We can see from the plot on Figure 2 (right) that random forest, gradient-boosted regression and support vector machines depict the best performance for smaller datasets. We show in line 209 the tendency of random forest and gradient-boosted regression to overfit in the case of smaller datasets. Thus we conclude that support vector machines are better-suited algorithms in case of fewer datasets. For clarity, we modify line 217 as follows: ***Figure 2b depicts the superior performance of RF, GBR and SVM in the event of limited dataset availability. However, we have seen that RF and GBR show a marked increase in training MAE with increasing training samples which suggests overfitting to limited datasets. Thus SVM is more robust to smaller datasets.***

**6 Reviewer Comment:** *L240: Instead of 'This is depicted in Fig 5.' just right '(Fig. 5).' at the end of the sentence*

**Author Response:** Thank you. We accept this suggestion.

**7 Reviewer Comment:** *L247: define somewhere in the manuscript what are 'ablation meteorological variables'*

**Author Response:** Thank you for this suggestion. We will incorporate an explanation of which variables contribute to ablation and accumulation in the data and methods section 2.2

**8 Reviewer Comment:** *L261: is 'cost' the same as 'penalty'? If so you should be consistent and use one or the other throughout the manuscript.*

**Author Response:** Thank you for bringing this to our notice. This is a remnant of an earlier iteration of manuscript preparation. We will correct all occurrences of this oversight.

**9 Reviewer Comment:** *L304: How do you conclude this ranking? From Figure 3 and Figure 4, it looks like RF and SVM are closer than SVM and NN.*

**Author Response:** Thank you for bringing this to our notice. We were interpreting using Figure 2. Here, a consistent shift in the performance of mean absolute error of random forest and SVM is evident. To improve clarity, we modify lines 303-305 to ***The GBR model resulted in the best testing performance MAE, RMSE and $R^2$ values outperforming the RF model, SVM and NN models. Neural networks resulted in better bias performance.***

**10 Reviewer Comment:** *L326: To which graphs are you linking here? Maybe include the figure number.*

**Author Response:** Thank you for bringing this to our notice. We meant the graph of LR model in Figure 2. We will specify the figure number to avoid confusion.

**11 Reviewer Comment:** *L348: Probably you could not find the expected importance of precipitation because it is poorly resolved in the climate input data (see GC2).*

**Author Response:** Thank you. This is likely. We will include the line ***This is***

150 ***possibly a result of the scale of the meteorological variables used.***

**12 Reviewer Comment:** *Table 1: Why is 'Number of trees' listed two times?*

**Author Response:** We apologize for the lack of clarity in the representation of the table. We have corrected the design of the table to include horizontal separators as depicted in Table 1:

Table 1: Grid of settings used for hyperparameter tuning of each of the models

| Machine learning model | Hyperparameter | Values |
|---|---|---|
| Random Forest | Number of trees | 10,20,50,100 |
| Gradient Boosted Regressor | Number of trees | 50,100,200 |
| | Subsampling | 0.7, 1.0 |
| | Maximum Depth | 3,5,10 |
| Support Vector Machine | Cost | 0.1, 1, 10, 20 |
| | Kernels | Sigmoid, Radial Basis Function, Polynomial |
| | Degree (polynomial kernel) | 2, 3, 4, 5 |
| Artificial Neural Network | Number of layers and nodes | **1:** 10, 50, 100, 200, 300, 400, 500, **2:** (100, 50), (200, 100), (400, 200), (200, 400) **3:** (400, 200, 100), (500, 200, 100), (200, 100, 50), (100, 50, 10), **4:** (200, 300, 400, 500), (300, 200, 100, 50), (200, 100, 50, 10) |

155 **13 Reviewer Comment:** *Figure 2: See GC4. In the caption also explain which quantiles are shown in the box plot on the left. And explain how the two plots are connected (are the yellow boxes on the left representing the quantiles of the lines in the right plot?) Add the unit to the y-axis. Currently wrong caption: 'Training and testing RMSE (in*

*mm we) and r values for varying the size of the training dataset for each of the models:'*

160 *but only shown is MAE.*

**Author Response:** Thank you for bringing this to our notice. We agree with the concerns raised. We will update the caption in line with this suggestion.

**14 Reviewer Comment:** *Figure 3: See GC4. In caption: e.g. how are training and testing data split in this plot,70%/30% or different, include (a), (b), (c) and (d) and* 165 *explain also in the caption which performance measure is shown in which subplot. Add units to the y-axis where needed.*

**Author Response:** Thank you for bringing this to our notice. We have corrected this figure to reflect the suggestions by the reviewer.

**15 Reviewer Comment:** *Figure 4: See GC4. Maybe you can include the information of* 170 *Figure 3 into this figure and delete Figure 3 (e.g. "RMSE: 0.95/1.08 mwe" and include a legend at the empty subplot space lower right with "RMSE: Training/Testing"). For the y-equations don't write y=0.744x + (-338.433) instead write y = 0.744x – 338.433. Is the high precision of numbers with three decimals meaningful for the RMA regression?*

**Author Response:** Thank you for this suggestion. In fact, the initial draft of this 175 manuscript included the information of Figure 3 in Figure 4 exactly as suggested by the reviewer and another iteration represented in the form of stacked line plots of all the metrics in the empty panel of Figure 4. However, both options appeared cluttered. To improve the readability a separate plot with the training and testing metric was included.

180 **16 Reviewer Comment:** *Figure 5: hard to distinguish in the legend what are the solid lines and what are the dashdotted lines. In the caption mention which test score is shown and explain briefly what the negative scaled RMSE is.*

**Author Response:** Thank you for bringing this to our notice. We will update the

legend and caption to reflect this suggestion.

**17 Reviewer Comment:** *Figure 6: In the caption mention which test score is shown and explain briefly what the negative scaled RMSE is.*

**Author Response:** Thank you. We will incorporate this suggestion. We have used the negative of the root mean squared error after scaling the target labels to a range between 0 and 1 as the test score. This makes the assigning of ranks to the hyperparameter combination setting more intuitive.

**18 Reviewer Comment:** *Figure 7: increase the font size, In Caption mention which test score is shown. Also include the test score name in the y-axis (currently only 'Test score').*

**Author Response:** Thank you for bringing this to our notice. We have corrected the font size and the test score details.

**19 Reviewer Comment:** *Maybe you could combine Figures 5, 6 and 7 into one Figure.*
**Author Response:** Thank you. Yes, we agree. Figures 5, 6 and 7 will be merged.

**20 Reviewer Comment:** *Figure 8: increase the font size. Because the x-axis is limited to 13 maybe add the numbers in the plot for features which go beyond this limit. Maybe include the abbreviations of meteorological variables in the caption or the text, so you can understand the plot without having a look in the supplementary. And you can also use the abbreviations in the result sections.*

**Author Response:** Thank you for your suggestion. To improve the representation of the feature importance and the meteorological variables without having to look at the supplementary file, we reformatted the image in the form of a RADAR plot. The tentative figure is as depicted in Figure **R2**.

**21 Reviewer Comment:** *Supplementary S1:*

[Figure]

Figure **R2**: Radar plot depicting the percentage importance of all features summed over the accumulation and ablation season for the models: Random Forest, Gradient Boosted Regression, Support Vector Machine, Artificial Neural Network and Linear Regression. The radial axis represents the summed percentage importance and the angular axis represents the input features.

- *general: give more meaningful names to the individual sheets*

- *sheet3: no explanation of what is shown on this sheet, include references in the text or delete this sheet*

**Author Response:** Thank you for bringing this to our notice. Sheet 3 will be deleted. Sheet 1 will also be deleted as the new Radar plot (Fig **R2**) contains the full name of the meteorological variables.

**References**

1. Radić, V., Bliss, A., Beedlow, A.C., Hock, R., Miles, E. and Cogley, J.G., 2014. Regional and global projections of twenty-first century glacier mass changes in response to climate scenarios from global climate models. Climate Dynamics, 42, pp.37-58.

2. Maussion, F., Butenko, A., Champollion, N., Dusch, M., Eis, J., Fourteau, K., Gregor, P., Jarosch, A.H., Landmann, J., Oesterle, F. and Recinos, B., 2019. The open global glacier model (OGGM) v1. 1. Geoscientific Model Development, 12(3), pp.909-931.

---

## Author Response (AR2)

**Modelling the Point Mass Balance for the Glaciers of Central European Alps using Machine Learning Techniques**

Ritu Anilkumar, Rishikesh Bharti, Dibyajyoti Chutia, Shiv Prasad Aggarwal
**Correspondance:** ritu.anilkumar@nesac.gov.in

We are grateful for the editor's thorough review and suggestions on manuscript number **egusphere-2022-1076**: 'Modelling the Point Mass Balance for the Glaciers of Central European Alps using Machine Learning Techniques'. Please find enclosed a marked-up version of the manuscript incorporating the suggested changes and the revised manuscript. The

5 point-by-point response to the comments is provided below. The comments by the editor are presented in a ***bold italicised*** font style. The author's response is in normal font style. Text quoted from the revised manuscript is *italicized*.
* * *
**COMMENT 1:**

***Could you please clarify what is meant by forecast albedo?***

Forecast albedo is a measure of the reflectivity of the surface. It is represented as a fraction of incident shortwave radiation that is reflected by the surface across the spectrum. It is different from the broadband albedo given by

$$1 - \frac{Net\ Solar\ Radiation}{Downward\ Solar\ Radiation}$$

. To clarify the importance of forecast albedo as observed using the machine learning modelling, we include the following at lines 389-393 (Section 4.3) in the revised manuscript:

10 *In the case of ERA5 Land, the forecast albedo variable represents both the direct and diffuse radiation incident on the surface with values dependent on the land cover type. It is calculated using a weight applied to the albedo in the UV-visible and infrared spectral regions. The*

*albedo of snow and ice land covers is different in the UV-visible spectral region and the infra-red spectral region. This makes forecast albedo more important than broadband albedo, which* depends only on the surface net solar radiation and the surface solar radiation downwards.*
* * *
**COMMENT 2:**

*Section 4.1: Although the focus of the paper is intercomparison of ML approaches, this section would benefit from the inclusion of available literature on ERA5-Land performance in complex terrain. Some relevant papers might be:*

- *https://doi.org/10.1016/j.jhydrol.2023.129384*

- *https://doi.org/10.3389/feart.2022.907730*

- *https://doi.org/10.1016/j.jhydrol.2021.127353*

- *https://www.sciencedirect.com/science/article/pii/S0169809520313028*

- *Specific to downscaling:*
  *https://agupubs.onlinelibrary.wiley.com/doi/full/10.1029/2021WR031294*

Thank you for this suggestion. We have updated Section 2.2 Data and Methods and Section 4.1 Comparison of Model Performance and Associated Errors to reflect this. The changes are in lines 140-145 in the revised manuscript (Section 2.2):

*"The network of weather stations is sparse over much of the Alpine terrain; hence, reanalysis datasets are recommended (Hersbach et al 2020). We used the ERA5-Land reanalysis dataset (Muñoz Sabater, 2019, 2021). This data set was chosen primarily due to its comparatively high spatial resolution. This is in line with the findings of Lin et al 2018 and Chen et al 2021 that suggest that datasets with higher spatial resolution effectively represent the orographic*

*drag and mountain valley circulation which in turn results in improved performance for oro-graphically complex terrain."*

The lines 335-343 are modified in the revised manuscript's Discussion Section 4.1 Comparison of Model Performance and Associated Errors.

*"Further, the use of input meteorological reanalysis data can result in bias, especially in locations without sufficient ground stations (Zandler et al 2019, Guidicelli et al 2022). Specifically for the use of ERA5 Land data in complex terrain, Wu et al 2023 reports that while ERA5 Land represents the intra-annual variations in precipitation characteristics, there is a positive bias in the precipitation variables. Similarly, in the case of temperature, Zhao et al 2022 show through correlation and RMSE analysis that while the ERA5 Land dataset captures the temperature trends effectively, the magnitude of the values is not well represented. Thus, we suggest using a bias correction step such as that proposed by Cucchi et al 2020 in the case of RF, GBR and SVM models. Moreover, the reanalysis data do not fully reflect point scale data as it has a coarse resolution. Lin et al 2018 depicts the impact of resolution in simulating drivers of local weather in complex terrain and shows that coarser resolutions do not account for orographic drag. "*
* * *
**COMMENT 3:**

***Section 4.3: Consistent with Reviewer 2's comments, I suggest mentioning the caveat that using only annual mass balance data could impact the analysis of feature importance for the accumulation and ablation months. I also suggest explicitly discussing what is known or expected about orographic precipitation in ERA5-Land as opposed to referring to data scale.***

We have incorporated the suggestion on orographic precipitation in lines 384-386 in the revised manuscript.

*"This is possibly a result of the scale of the meteorological variables used not sufficiently*

*representing the influence of orographic water vapour transport that results in precipitation (Lin et al 2018, Chen et al 2021)."*

The suggestion on mentioning the caveat of using annual mass balance datasets is accepted and presented in Section 4.4 (lines 434-437) of the revised manuscript.

*" An important factor to note is that through this study, we have considered annual mass balance measurements as opposed to seasonal measurements due to the paucity of sufficient datasets to fully train a multi-parameters machine learning model. The role of ablation and accumulation variables will be better represented in the case of seasonal measurements and is an avenue to explore through future studies."*
* * *
**COMMENT 4:**

**Please provide a more descriptive name for the supplementary sheet as well as a small caption for the data that are presented. I suggest also mentioning any sensitivity studies that were performed (e.g., using Leaky ReLU) and their results in the supplement.**

A revised supplementary zip file with a text file containing captions is uploaded to https://github.com/RituAnilkumar/pt-gmb-ml
* * *
**References**

1. Lin, C., Chen, D., Yang, K. and Ou, T., 2018. Impact of model resolution on simulating the water vapor transport through the central Himalayas: implication for models' wet bias over the Tibetan Plateau. Climate dynamics, 51, pp.3195-3207.

2.

3. Zandler, H., Haag, I. and Samimi, C., 2019. Evaluation needs and temporal performance differences of gridded precipitation products in peripheral mountain regions. Scientific reports, 9(1), pp.1-15.

4. Cucchi, M., Weedon, G.P., Amici, A., Bellouin, N., Lange, S., Müller Schmied, H., Hersbach, H. and Buontempo, C., 2020. WFDE5: bias-adjusted ERA5 reanalysis data for impact studies. Earth System Science Data, 12(3), pp.2097-2120.

5. Chen, Y., Sharma, S., Zhou, X., Yang, K., Li, X., Niu, X., Hu, X. and Khadka, N., 2021. Spatial performance of multiple reanalysis precipitation datasets on the southern slope of central Himalaya. Atmospheric Research, 250, p.105365.

6. Muñoz-Sabater, J., Dutra, E., Agustí-Panareda, A., Albergel, C., Arduini, G., Balsamo, G., Boussetta, S., Choulga, M., Harrigan, S., Hersbach, H. and Martens, B., 2021. ERA5-Land: A state-of-the-art global reanalysis dataset for land applications. Earth System Science Data, 13(9), pp.4349-4383.

7. Guidicelli, M., Huss, M., Gabella, M. and Salzmann, N., 2023. Spatio-temporal reconstruction of winter glacier mass balance in the Alps, Scandinavia, Central Asia and western Canada (1981–2019) using climate reanalyses and machine learning. The Cryosphere, 17(2), pp.977-1002.

8. Zhao, P. and He, Z., 2022. A first evaluation of ERA5-Land reanalysis temperature product over the Chinese Qilian Mountains. Frontiers in Earth Science, 10, p.907730.

9. Wu, X., Su, J., Ren, W., Lü, H. and Yuan, F., 2023. Statistical comparison and hydrological utility evaluation of ERA5-Land and IMERG precipitation products on the Tibetan Plateau. Journal of Hydrology, 620, p.129384.